# Ensemble of Averages: Improving Model Selection and Boosting Performance in Domain Generalization

**Devansh Arpit, Huan Wang, Yingbo Zhou, Caiming Xiong**
Salesforce Research, USA
devansharpit@gmail.com

## Abstract

In Domain Generalization (DG) settings, models trained independently on a given set of training domains have notoriously chaotic performance on distribution shifted test domains, and stochasticity in optimization (e.g. seed) plays a big role. This makes deep learning models unreliable in real world settings. We first show that this chaotic behavior exists even along the training optimization trajectory of a single model, and propose a simple model averaging protocol that both significantly boosts domain generalization and diminishes the impact of stochasticity by improving the rank correlation between the in-domain validation accuracy and out-domain test accuracy, which is crucial for reliable early stopping. Taking advantage of our observation, we show that instead of ensembling unaveraged models (that is typical in practice), ensembling moving average models (EoA) from independent runs further boosts performance. We theoretically explain the boost in performance of ensembling and model averaging by adapting the well known Bias-Variance trade-off to the domain generalization setting. On the DomainBed benchmark, when using a pre-trained ResNet-50, this ensemble of averages achieves an average of $68.0\%$, beating vanilla ERM (w/o averaging/ensembling) by $\sim 4\%$, and when using a pre-trained RegNetY-16GF, achieves an average of $76.6\%$, beating vanilla ERM by $6\%$. Our code is available at https://github.com/salesforce/ensemble-of-averages.

## 1   Introduction

Domain generalization (DG, [5]) aims at learning predictors that generalize well on data sampled from test distributions that are different from the training distribution. Currently, deep learning models have been shown to be poor at this form of generalization [10], and excel primarily in the IID setting [51].

While a number of algorithms have been proposed to mitigate this problem (cf [51] for a survey), [18] demonstrate that models trained using empirical risk minimization (ERM, [43]) along with proper model selection (i.e. early stopping using validation set), using a subset of data from all the training domains, largely match or even outperform the performance of most existing domain generalization algorithms. This suggests that model selection plays an important role in domain generalization. Despite its importance, *there has not been much investigation into the reliability of model selection.* As we demonstrate in Figure 1, the out-domain performance varies greatly along the optimization trajectory of a model during training, even though the in-domain performance does not. This instability therefore hurts the reliability of model selection, and can become a problem in realistic settings where test domain data is unavailable, because it causes the rank correlation between in-domain validation accuracy and out-domain test accuracy to be weak.

In this paper, we first investigate a simple protocol for model averaging that both boosts DG within the ERM framework, and mitigates performance instability of deep models on out-domain data,

36th Conference on Neural Information Processing Systems (NeurIPS 2022).

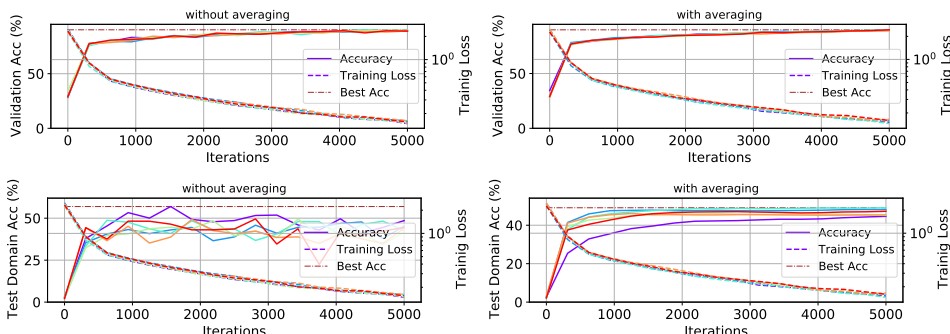

Figure 1: Model averaging improves out-domain performance *stability*. **Left**: In-domain validation accuracy and out-domain test accuracy during training of models using ERM. **Right**: Same as left, except validation and test predictions are made using a simple moving average of the model being optimized, along its optimization path. **Details**: The plots are for the TerraIncognita dataset with domain L38 used as the test domain, and others as training/validation data, and ResNet-50. Solid lines denote accuracy, dashed lines denote training loss, and dash-dot lines denote best accuracy achieved during training and all runs (for reference). Each color denotes a different run with a different random seed and training/validation split. **Gist**: Model averaging reduces out-domain performance instability, and makes the test curves correlate better with the validation curves, making model selection using in-domain validation set more reliable during optimization. We see a similar pattern when using ensemble of models, with and without model averaging, in Figure 2.

specifically with respect to in-domain validation data. This makes model selection more reliable. Next, taking advantage of our observation, we show that ensembling moving average models further boosts performance, making it a better choice for practical scenarios. Note that we do not claim that model averaging or ensembling can fully solve the problem of DG. The observation that model averaging can boost domain generalization performance is not new, and was exposed by SWAD [8], which inspired our work. Our contribution in this respect are as follows:

1. **Hyperparameter-free**:In contrast to SWAD, which introduces three additional hyper-parameters for its model averaging algorithm that need tuning, we show that the simple strategy of maintaining a simple moving average (SMA) of the model parameters throughout the optimization trajectory, starting near initialization (Appendix Figure 5), works just as well (when a pre-trained model is used as initialization). Although model averaging technically requires two hyper-parameters– averaging frequency and starting iteration, through empirical analysis, we show that setting the frequency to 1 and setting the start iteration close to 0 works well on multiple datasets and architectures, making our proposal hyperparameter-free in practice.

2. **Computationally efficient**:SWAD requires computing validation performance more frequently than is typically done (2x-6x on the DomainBed datasets), which is needed because it needs to find the start and end iteration between which model averaging is done. This increases compute requirements. This segment is selected based on the validation performance computed using the model being trained. Our proposal to instead use the SMA model to perform early stopping and inference, side-steps this need and does not require frequent validation performance check. We show that the root cause for this difference is that the model being trained has unstable performance on OOD data, while the SMA model has a more stable OOD performance (see Figure 1 and Table 2). Thus this observation results in our hyperparameter-free and more efficient model averaging strategy.

3. **EoA**: Taking advantage of our efficient model averaging protocol (section 2.2), we find that an ensemble of moving average models (EoA) outperforms a traditional ensemble of unaveraged models (Table 4). We also show ablation analysis that the rank correlation between in-domain validation performance and out-domain test performance is better for the ensemble of average models (Table 3).

4. **Theoretical explanation**: To explain why both model averaging and ensembling improve OOD performance under a unified theoretical framework, we adapt the well known Bias-Variance decomposition to the domain generalization setting, and argue that the expected OOD loss for individual models comprises of both the bias and the variance term, while the expected OOD loss for ensembles and averaged models comprises mainly of the bias term only, and is thus strictly lower (section 3.2). Our explanation is in contrast with SWAD, which uses flat minima to explain the improved OOD

generalization, which applies to model averaging, but is less straight forward for explaining the boost by ensembles.

5. **Benchmarking**: For benchmarking, we experiment with three different pre-trained models as initializations for DG training, with increasing pre-training dataset size and model size. In these experiments we find that EoA provides a larger gain over the *corresponding* ERM baseline with increasing dataset and model size. These gains range from $4\% - 6\%$ (Table 4). Notice that this claim is different from existing work [20], which states that the baseline ERM performance improves with larger pre-training data and model size.

## 2 Model Averaging

### 2.1 Terminology

**Online Model**: For a given supervised learning objective function, let $f_\theta(.)$ denote the deep network being optimized using gradient based optimizer, where $\theta$ denotes the parameters of this model. We refer to $f_\theta$ as the *online model*, or *unaveraged model*. The output of $f_\theta(.)$ is a vector of $K$ logits corresponding to the $K$ classes in the supervised task.

**Moving Average (MA) Model**: While the online model is being trained, we maintain a moving average of the online model's parameters. This process is sometime referred to as *iterate averaging* in existing literature. The deep network whose parameters are set to be this moving average is referred to as the *moving average model*, or more specifically *simple moving average (SMA) model* because of its use in our work. We denote the parameters of this model by $\hat{\theta}$.

### 2.2 Model Averaging Protocol

We use a simple moving average (SMA) of the online model. Instead of calculating the moving average starting from initialization (as done in Polyak-Ruppert averaging), we instead start after a certain number of iterations $t_0$ during training (tail averaging), and maintain the moving average until the end of training. As we discuss in the next section, $t_0$ is chosen to be close, but not equal to the initialization when a pre-trained model is used as initialization. At any iteration $t$, we denote:

$$\hat{\theta}_t = \begin{cases} \theta_t, & \text{if } t \leq t_0 \\ \frac{t-t_0}{t-t_0+1} \cdot \hat{\theta}_{t-1} + \frac{1}{t-t_0+1} \cdot \theta_t, & \text{otherwise} \end{cases} \tag{1}$$

where $\theta_t$ is the online model's state at iteration $t$. Note that effectively, $\hat{\theta}_t := \frac{1}{t-t_0+1} \cdot \sum_{t'=t_0}^t \theta_{t'}$. Further, at iteration $t$, if we need to calculate validation performance, we use $\hat{\theta}_t$ to do so, and not $\theta_t$. As we show in the next section, the benefit of doing so is that the rank correlation between in-domain validation accuracy and out-domain test accuracy is significantly better when predictions are made using $\hat{\theta}_t$. This makes model selection more reliable for domain generalization. Finally, for a given run, model selection selects $\hat{\theta}_{t*}$ for making test set predictions, such that $\hat{\theta}_{t*}$ achieves the best validation performance. We discuss some theoretical perspectives on why model averaging can help domain generalization in section 5.1.

### 2.3 Ablation Analysis

Here we perform four ablation studies: 1) impact of the start iteration $t_0$ used in our SMA protocol in Eq. 1; 2) the frequency of model averaging; 3) instability reduction of SMA model compared to the online mode along the optimization trajectory on out-domain data; 4) correlation between in-domain and out-domain accuracy across independently trained models.

Due to space limitation, we show experiments for 1,2 and 4 in Appendix section C. In summary, we find that: 1) starting averaging close to initialization results in improved out-domain performance (Figure 5 in Appendix) when the parameters are initialized used a pre-trained model; 2) the frequency of SMA does not have a significant impact on performance, unless sampling is done at too large intervals (Figure 6 in Appendix); 4) the rank correlation is poor between validation and test accuracy of *independently* trained models (Figure 8 in Appendix). An implication of this is that it is difficult to discover the best model (for out-domain performance) from a pool of independently trained models, based only on their in-domain validation performance (echoing the findings of [10]).

Table 1: Spearman correlation (closer to 1 is better) between within-run in-domain validation accuracy and out-domain test accuracy on multiple datasets. Model averaging improves rank correlation for both individual models (left) and ensemble of averages (right).

<table>
<tr><td colspan="3" align="center">Table 2: Individual Models</td><td colspan="3" align="center">Table 3: Ensembles</td></tr>
<tr><td>TerraIncognita</td><td>w/o avg</td><td>w/ avg</td><td>TerraIncognita</td><td>w/o avg</td><td>w/ avg</td></tr>
<tr><td>L100</td><td>$0.21 \pm 0.07$</td><td>$\mathbf{0.90 \pm 0.05}$</td><td>L100</td><td>0.48</td><td>**1**</td></tr>
<tr><td>L38</td><td>$0.12 \pm 0.13$</td><td>$\mathbf{0.83 \pm 0.05}$</td><td>L38</td><td>0.17</td><td>**0.95**</td></tr>
<tr><td>L43</td><td>$0.30 \pm 0.06$</td><td>$\mathbf{0.67 \pm 0.18}$</td><td>L43</td><td>**0.59**</td><td>0.38</td></tr>
<tr><td>L46</td><td>$0.03 \pm 0.11$</td><td>$\mathbf{0.52 \pm 0.14}$</td><td>L46</td><td>0.08</td><td>**0.61**</td></tr>
</table>

### 2.3.1 Instability Reduction: Rank Correlation

We study the reliability of model selection for domain generalization when using online models vs moving average models, using rank correlation (see Appendix C.4 for definition). To do so, we train models on a dataset, both with and without model averaging, and compute Spearman correlation between the in-domain validation accuracy and out-domain test accuracy sampled at regular intervals during the training process. Since there are multiple runs where a given domain acts as the test domain, we calculate the mean and standard error of these values over these runs.

The rank correlations are shown in Table 2 (and Table 8 in Appendix) for the PACS, VLCS, Office-Home, TerraIncognita and DomainNet datasets. We find that in majority of the cases, using model averaging results in a significantly better rank correlation compared to using the online model. These experiments therefore suggest that the reliability of model selection is significantly higher within a run when using model averaging.

## 3 Ensemble of Averages (EoA)

[18] propose a rigorous framework for evaluation in the domain generalization setting which accounts for randomness due to seed and hyper-parameter values, and recommend reporting the average test accuracy over all the runs computed using a model selection criteria. However, in practice, it is desirable to have a single predictor that has a high accuracy. An ensemble combines predictions from multiple models, and is a well known approach for achieving this goal [11] by exploiting function diversity [14]. However, as we show, even ensembles suffer from instability in the domain generalization setting. Building on the observations of the previous section, we investigate the behavior of ensemble of moving average models and find that it mitigates this issue. We begin by describing the EoA protocol below.

**EoA Protocol**: We perform experiments with ensemble of multiple independently trained models (i.e., with different hyper-parameters and seeds). When each of these models are moving average models from their corresponding runs, we refer to this ensemble in short as the *ensemble of averages (EoA)*. Identical to how we make predictions for traditional ensembles (specifically the bagging method [6]), the class $\hat{y}$ predicted by an EoA for an input $\mathbf{x}$ is given by the formula:

$$\hat{y} = \arg\max_k Softmax(\frac{1}{E}\sum_{i=1}^{E} f(\mathbf{x}; \hat{\theta}_i))_k \qquad (2)$$

where $E$ is the total number of models in the ensemble, $\hat{\theta}_i$ denotes the parameters of the $i^{th}$ moving average model, and the sub-script $(.)_k$ denotes the $k^{th}$ element of the vector argument. Finally, the state $\hat{\theta}_i$ of the $i^{th}$ moving average model used in the ensemble is selected from its corresponding run using its in-domain validation set performance (described in section 2.2). We now investigate the behavior of EoA compared with ensembles of online models on domain generalization tasks.

### 3.1 Analysis

**Qualitative visualization**: For the purpose of contrasting the behavior of traditional ensembles vs ensemble of averages, we begin by qualitatively studying the stability of out-domain performance of these two ensembling techniques during the training process. To do so, we use the TerraIncognita

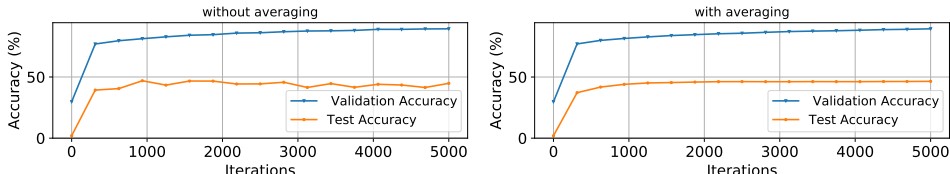

Figure 2: Ensemble of moving averages (EoA) (right) has better out-domain test performance *stability* compared with ensemble of online models (left), w.r.t. in-domain validation accuracy. **Details**: The plots are for the TerraIncognita dataset with domain L38 used as the test domain, and others as training/validation domain, and ResNet-50. Each ensemble has 6 different models from independent runs with different random seeds, hyper-parameters, and training/validation split.

dataset, and fix one of its domains as the test domain while using the others as training/validation data. We then train 6 different models independently for $5,000$ iterations with different seeds, hyper-parameters and training-validation splits identical to the [18] protocol. We also maintain moving average models corresponding to each of these 6 models. At every $300$ iterations, we form an ensemble of the 6 online models from their corresponding runs and compute the out-domain test accuracy. Since, each run has a different training-validation split, we calculate the mean validation accuracy of each of these online models at that iteration. We follow an identical procedure for the moving average models and plot these performances in Figure 2. We find that the ensemble of averages has a better stability on out-domain test set compared to the ensemble of online models.

For clarity, note that this procedure for calculating test accuracy at regular intervals is different from what we proposed earlier for EoA for practical purposes. This experiment is only meant to highlight the fact that making predictions on out-domain data using an ensemble of online models suffers from instability along the optimization trajectory, while an ensemble of averages mitigates this issue. For plots on other domains of TerraIncognita, see Figure 10 in the Appendix.

**Rank correlation**: We now measure the rank correlation between in-domain validation accuracy and out-domain test accuracy for a quantitative evaluation. The details of the metric and motivations behind this experiment are same as those described in section 2.3.1. Here we use the same experimental setup described in the qualitative analysis above. But in addition, we also conduct experiments on VLCS, OfficeHome and DomainNet datasets. The results are shown in Table 3 (and Table 9 in Appendix). We find that in majority of the cases, using EoA results in a significantly better rank correlation compared to using the online model ensemble. These results show more concretely the fact that predictions by an ensemble of online models on out-domain data suffers from instability along the optimization trajectory, and EoA mitigates this problem.

## 3.2   Why does Ensembling and Model Averaging Improve Performance?

We explain the performance boost achieved by ensemble of averages (see next section) by adapting the Bias-Variance decomposition [17] to the domain generalization setting. For classification tasks with one-hot labels, the Bias-Variance decomposition is given as [49],

$$\mathbb{E}_{\mathbf{x},y}\mathbb{E}_{\mathcal{T}}[CE(y, f(\mathbf{x};\mathcal{T}))] = \underbrace{\mathbb{E}_{\mathbf{x},y}[CE(y, \bar{f}(\mathbf{x}))]}_{\text{Bias}^2} + \underbrace{\mathbb{E}_{\mathbf{x},\mathcal{T}}[KL(\bar{f}(\mathbf{x}), f(\mathbf{x};\mathcal{T}))]}_{\text{Variance}}$$

where $CE$ denotes the cross entropy loss, $KL$ denotes KL divergence, $\mathcal{T} = \{(\mathbf{x}_i^{in}, y_i^{in})\}_{i=1}^{N}$ are $N$ IID samples drawn from the in-domain training distribution $\mathbb{P}^{in}$, $f(\mathbf{x};\mathcal{T})$ denotes the prediction of the model $f$ on sample $\mathbf{x}$ such that the model is trained on the dataset $\mathcal{T}$, and $\bar{f}(\mathbf{x}) = \mathbb{E}_{\mathcal{T}}[f(\mathbf{x};\mathcal{T})]$. Finally $(\mathbf{x}, y) \sim \mathbb{P}^{out}$ where $\mathbb{P}^{out}$ is the out-domain distribution. Notice how $\mathcal{T}$ and $(\mathbf{x}, y)$ come from different distributions. For instance, in PACS dataset, $\mathbb{P}^{in}$ could be the union of art, cartoon and photo domains, and $\mathbb{P}^{out}$ could be the sketch domain.

The L.H.S. of the above equation is the expected cross entropy loss on the out-domain distribution achieved by individual models, i.e., when we train an individual model on a particular instance of the training dataset $\mathcal{T}$, the expected out-domain test loss is denoted by L.H.S. Importantly, the Bias term on the R.H.S. denotes the expected cross entropy loss on the out-domain distribution achieved by the function $\bar{f}(.)$, which is essentially an *ensemble*. Finally, the variance term captures how much the

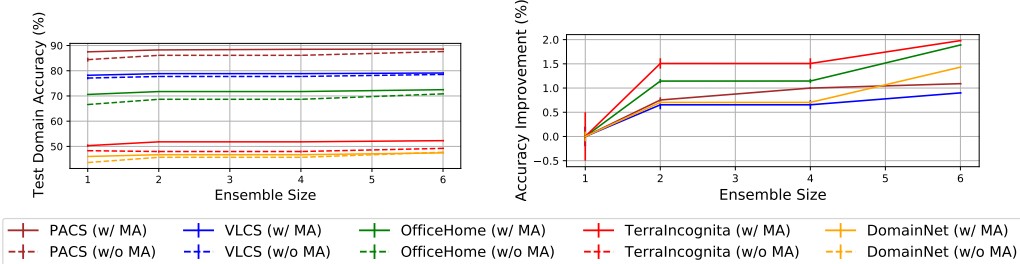

Figure 3: **Left**: Effect of ensemble size (number of models in an ensemble) on out-domain performance (mean and standard error) for models with and without moving average (MA) parameters for ResNet-50 pre-trained on ImageNet. **Right**: Using the performance of ensemble of size 1 (shown in the left plot) as reference, right plot shows the percentage point improvement for ensembles of size $> 1$. The plots show that i) ensemble of averages (solid lines in left plot) are consistently better than ensemble of models without averaging (dashed lines in left plot); ii) ensemble of averages consistently improves performance over averaged models (ensemble of size 1 in right plot).

prediction of individual models differs in expectation from the ensemble prediction, which makes this term strictly greater than zero.

Therefore, the above decomposition tells us that the expected test domain error of an ensemble is strictly less than that of an individual model. This interpretation directly explains why a traditional ensemble of unaveraged models can be expected to perform better than individual unaveraged models. However, it is still not clear why EoA performs better that a traditional ensemble in practice. To establish this connection, we note that in practice, we typically train a small number of independent models to form a traditional ensemble due to computational constraints. Thus such ensembles do not behave identically to the expected ensemble $\bar{f}(.)$ described above. Model averaging on the other hand has been shown to approximate an ensemble [23]. To see this, consider without any loss of generality that the ensemble contains models with parameters $\{\theta_1, \theta_2 \dots \theta_T\}$, and denote $\hat{\theta}_T := \frac{1}{T} \cdot \sum_{t=1}^{T} \theta_t$. Then note that the second order Taylor's expansion around $\hat{\theta}_T$ of each model's $k^{th}$ dimension's prediction is given by,

$$\frac{1}{T} \cdot \sum_{t=1}^{T} f(\theta_t)_k \approx f(\hat{\theta}_T)_k + \frac{1}{T} \cdot \sum_{t=1}^{T} (\hat{\theta}_T - \theta_t)^T \frac{\partial f(\hat{\theta}_T)_k}{\partial \hat{\theta}_T} + 0.5 (\hat{\theta}_T - \theta_t)^T \frac{\partial^2 f(\hat{\theta}_T)_k}{\partial \hat{\theta}_T^2} (\hat{\theta}_T - \theta_t)$$

Notice that $f(.)$ is the model output and therefore the first and second order terms are the derivatives of the model output and not the loss gradient and Hessian. The first order term is zero due to $\hat{\theta}_T := \frac{1}{T} \cdot \sum_{t=1}^{T} \theta_t$. A crucial difference of our analysis compared to [23] is that they average model states that lie near different loss minima, while we perform tail averaging. Therefore, the term $(\hat{\theta}_T - \theta_t)$ may not behave similar to that in their case. To shed light on its behavior, we plot the histogram of the second order term and the moving average model's logit $f(\hat{\theta}_T)_k$ in Eq. 3 for the first dimension ($k = 1$) for test domain data in figure 4 (details and additional experiments provided in Appendix D). The histogram shows that the second order term

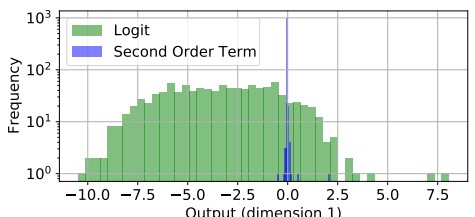

Figure 4: The scale of terms– moving average model's logit and the second order term in Eq. 3. The latter concentrates around 0, suggesting our model averaging protocol approximates ensembles.

concentrates near zeros while the logit values span a wider range, which implies that under the second order approximation, the model averaging protocol used in our work behaves like an ensemble. Finally, to study the impact of ensemble size on out-domain performance, we plot the test domain accuracy as a function of ensemble size in figure 3. The plots show that i. EoA outperforms traditional ensembles for all ensemble sizes (left); and ii. ensembles of larger size typically have better

Table 4: Performance benchmarking on 5 datasets of the DomainBed benchmark using two different pre-trained models. SWAD and MIRO are the previous SOTA. **See Table 10 in Appendix for comparison with more methods.** Note that ensembles do not have confidence interval because an ensemble uses all the models to make a prediction. Gray background shows our proposal. *Our runs* implies we ran experiments, but we did not propose it. Experiments use the training-domain validation protocol from [18].

| Algorithm | PACS | VLCS | OfficeHome | TerraIncognita | DomainNet | Avg. |
|---|---|---|---|---|---|---|
| ResNet-50 (25M Parameters, Pre-trained on ImageNet) | | | | | | |
| ERM (our runs) | $84.4 \pm 0.8$ | $77.1 \pm 0.5$ | $66.6 \pm 0.2$ | $48.3 \pm 0.2$ | $43.6 \pm 0.1$ | 64.0 |
| Ensemble (our runs) | 87.6 | 78.5 | 70.8 | 49.2 | **47.7** | 66.8 |
| ERM [18] | $85.7 \pm 0.5$ | $77.4 \pm 0.3$ | $67.5 \pm 0.5$ | $47.2 \pm 0.4$ | $41.2 \pm 0.2$ | 63.8 |
| SWAD [8] | $88.1 \pm 0.4$ | $\mathbf{79.1 \pm 0.4}$ | $70.6 \pm 0.3$ | $50.0 \pm 0.4$ | $46.5 \pm 0.2$ | 66.9 |
| MIRO [9] | $85.4 \pm 0.4$ | $79.0 \pm 0.$ | $70.5 \pm 0.4$ | $50.4 \pm 1.1$ | $44.3 \pm 0.2$ | 65.9 |
| SMA (ours) | $87.5 \pm 0.2$ | $78.2 \pm 0.2$ | $70.6 \pm 0.1$ | $50.3 \pm 0.5$ | $46 \pm 0.1$ | 66.5 |
| EoA (ours) | **88.6** | **79.1** | **72.5** | **52.3** | 47.4 | **68.0** |
| ResNeXt-50 32x4d [48] (25M Parameters, Pre-trained 1B Images) | | | | | | |
| ERM (our runs) | $88.9 \pm 0.3$ | $79.0 \pm 0.1$ | $70.9 \pm 0.5$ | $51.4 \pm 1.2$ | $48.1 \pm 0.2$ | 67.7 |
| Ensemble (our runs) | 91.2 | 80.3 | 77.8 | 53.5 | 52.8 | 71.1 |
| SMA (ours) | $92.7 \pm 0.3$ | $79.7 \pm 0.3$ | $78.6 \pm 0.1$ | $53.3 \pm 0.1$ | $53.5 \pm 0.1$ | 71.6 |
| EoA (ours) | **93.2** | **80.4** | **80.2** | **55.2** | **54.6** | **72.7** |
| RegNetY-16GF [40] (81M Parameters, Pre-trained on 3.6B Images) | | | | | | |
| ERM (our runs) | $92 \pm 0.4$ | $78.6 \pm 0.6$ | $73.8 \pm 0.5$ | $55.6 \pm 0.9$ | $53.1 \pm 0.2$ | 70.6 |
| Ensemble (our runs) | 95.1 | 80.6 | 80.5 | 59.5 | 57.8 | 74.7 |
| ERM [9] | $89.6 \pm 0.4$ | $78.6 \pm 0.3$ | $71.9 \pm 0.6$ | $51.4 \pm 1.8$ | $48.5 \pm 0.6$ | 68.0 |
| SWAD [9] | $94.7 \pm 0.2$ | $79.7 \pm 0.2$ | $80.0 \pm 0.1$ | $57.9 \pm 0.7$ | $53.6 \pm 0.6$ | 73.2 |
| MIRO [9] | $\mathbf{97.4 \pm 0.2}$ | $79.9 \pm 0.6$ | $80.4 \pm 0.2$ | $58.9 \pm 1.3$ | $53.8 \pm 0.1$ | 74.1 |
| SMA (ours) | $95.5 \pm 0.0$ | $80.7 \pm 0.1$ | $82.0 \pm 0.0$ | $59.7 \pm 0.0$ | $60.0 \pm 0.0$ | 75.6 |
| EoA (ours) | 95.8 | **81.1** | **83.9** | **61.1** | **60.9** | **76.6** |

out-domain performance. See a discussion on functional diversity of ensembles vs model averaging in Appendix E.

# 4  Empirical Results

## 4.1  DomainBed Benchmarking

We now benchmark our model averaging protocol (SMA) and ensemble of averages against online models (ERM, without MA) and ensemble of online models (ensembles). Note that all these models are trained using the ERM objective as before. We evaluate on PACS [27], VLCS [13], OfficeHome [45], TerraIncognita [3] and DomainNet [35] datasets in DomainBed. The training-evaluation protocols are the same as described in section 2.3 for moving average and online models, and in section 3 for ensembles. Full details can be found in section B in the Appendix.

**Comparison with existing results using ResNet-50 pre-trained on ImageNet**: Here we compare existing methods with our runs. All methods use ResNet-50 (25M parameters) [19] pre-trained on ImageNet as initialization. Comparing ERM [18] and ERM (our runs), we find that they perform similarly, especially considering we have used a smaller hyper-parameter space (further discussion in Appendix E). A comparison between SWAD and SMA shows that SWAD is slightly better (by $0.4\%$ on average). However, recall that our protocol retains the advantage of not tuning any hyper-parameters while SWAD has 3 additional ones that they tune separately in addition to the optimization hyper-parameters. Interestingly, traditional ensembles and SMA achieve similar performance ($66.8\%$ and $66.5\%$ respectively). Finally, EoA outperforms all the existing results: ERM by $4\%$ and SWAD (previous SOTA) by $1.1\%$. Importantly, note that while all non-ensemble models report the average test accuracy of multiple models following the protocol of [18], EoA test accuracy is achieved by a single predictor that combines the output of multiple models.

**Experiments with larger pre-training datasets and larger models**: In addition to ResNet-50 pre-trained on ImageNet, we now also experiment with ResNeXt-50 32x4d (25M parameters), that is pre-trained using semi-weakly supervised objective on Instagram 1B images and ImageNet labeled data [48], and RegNetY-16GF (81M parameters) pre-trained using Instagram 3.6B images. Note that both ResNet-50 and ResNeXt-50 32x4d have similar number of parameters, while RegNetY-16GF has more than 3x the number of parameters. On the other hand, also notice that the three architectures

are respectively pre-trained on an increasing size of datasets. The rationale behind this choice is that recent trends in deep learning has shown that models pre-trained on larger datasets and architectures achieve better downstream transfer performance [12, 32, 20]. Therefore, we expect the latter models to improve the ERM baseline, and our goal is to investigate the out-domain performance gain by model averaging and EoA relative to the *corresponding* ERM baseline with increasing pre-training dataset size and model size.

The experimental results are shown in Table 4. To investigate models with the same size, but one pre-trained on a larger dataset, we compare the results of ResNet-50 and ResNeXt-50 32x4d. On average across all five datasets, the gain of SMA over ERM (our runs) is $2.5\%$ for ResNet-50 and $3.9\%$ for ResNeXt-50 32x4d. The gain of EoA over ERM is larger: $4\%$ vs $5\%$ respectively. This suggests that pre-training the model on a larger dataset increases the gain of model averaging and EoA over the *corresponding* ERM baseline, while the ERM performance itself improves.

Next, to investigate the impact of both larger model size and larger pre-training dataset, we compare the results of ResNeXt-50 32x4d and RegNetY-16GF. On average across all five datasets, the gain of SMA over ERM (our runs) is $3.9\%$ for ResNeXt-50 32x4d and $5\%$ for RegNetY-16GF. The gain of EoA over ERM is again larger: $5\%$ vs $6\%$ respectively. This suggests that increasing both model size and pre-training dataset size allow model averaging and EoA to provide larger out-domain gains over the corresponding ERM baseline. Notice that these claims are different from existing work [20], which states that the baseline ERM performance improves with larger pre-training data and model size.

### 4.2 In-domain Performance Improvement using Model Averaging

We study the in-domain test accuracy on PACS and OfficeHome datasets using ImageNet pre-trained ResNet-50 with and without our SMA protocol. In this experiment, we combine all the domains of PACS and split it into training/-validation/test splits (0.8/0.1/0.1). We run 10

Table 5: SMA outperforms ERM without model averaging in the IID setting.

| Algorithm | PACS | OfficeHome |
|---|---|---|
| ERM (no averaging) | $94.39 \pm 0.46$ | $77.09 \pm 0.57$ |
| SMA (ours) | $\mathbf{96.77 \pm 0.20}$ | $\mathbf{83.56 \pm 0.21}$ |

different runs with different seeds and randomly chosen splits for each dataset. The best model for each run is chosen using the validation set. The remaining optimization details are identical to those used in the previous section. The test accuracy mean and standard error using these best models are shown in Table 5. As expected, SMA outperforms models without averaging.

## 5 Related Work

### 5.1 Model Averaging

**A theoretical perspective**: In our model averaging protocol, we compute a simple moving average of the model parameters starting early during training. This is known as *tail-averaging* [24], which is slightly different from Polyak-Ruppert averaging [36] in that the latter starts averaging from the very beginning of training. In the context of least square regression in the IID setting, [24] theoretically study the behavior of tail averaging and show that the excess risk of the moving average model is upper bounded by a bias and a variance term. This bias term depends on the initialization state of the parameter, but interestingly, it decays exponentially with $t_0$, where $t_0$ is the iteration at which model averaging is started. The variance term on the other hand depends on the covariance of the noise inherent in the data w.r.t. the optimal parameter, and is shown to decay at a faster rate when using model averaging, as opposed to a slower rate without averaging. This motivated them to propose *tail-averaging*.

Model averaging has also been shown to have a regularization effect [34] similar to that of Tikhonov regularization [42]. This regularization has been classically used in ill-posed optimization problems (typically least squared regression), which are *under-specified*. This property provides an interesting connection between model averaging and the *under-specification* problem discussed in [10], where the authors perform large scale experiments showing that the performance of multiple over-parameterized deep models, trained independently with different hyper-parameters and seeds, have a high variance on out-domain data, even though their in-domain performances are very close together. Based on this connection, a simple intuition why one can expect model averaging to help in domain generalization is its Tikhonov regularization effect. However, this intuition requires a more thorough investigation.

**SWAD** [8]: SWAD propose flat minima as a means for improving domain generalization. Following the intuition of stochastic weight averaging (SWA, [23]), they use model averaging to find flat minima. However, their proposal is different from sampling model states at regular intervals and towards the end of training (as done in SWA). SWAD selects contiguous model states along the optimization path for averaging, based on their validation loss. This is done to prevent including an under-performing state (determined using the in-domain validation set) in the moving average model. SWAD however adds additional hyper-parameters of its own: the validation loss threshold below which the the model states are selected, and patience parameters (number of iterations that determine the start and end of the averaging process). Note that this also requires computing validation loss more frequently during training. In this context, we show that instead of finding the start and end period for model averaging meticulously, we can simply start model averaging early during training and continue till the end. This difference arises from the fact that SWAD uses the online network to calculate validation performance while we use the SMA model in our protocol. This is explained further in section 2.2. The benefit our observations provide over SWAD is that they allow us to take advantage of model averaging without the additional hyper-parameters and compute required by SWAD.

## 5.2 Domain Generalization

Existing methods aimed at domain generalization can be broadly categorized into techniques that perform domain alignment, regularization, data augmentation, and meta-learning. Domain alignment is perhaps the most intuitive direction, in which methods aim to learn latent representations which have similar distributions across different domains [41, 30, 39, 37]. There are different variants of this idea, such as minimizing some divergence metric between the latent representation of different domains (E.g. DANN [16]), or less strictly, minimizing the difference between the latent statistics of different domains (E.g. DICA [33], CORAL [41]). In the meta learning category, source domains are typically split into 2 subsets to be used as the training and test domains in episodes to simulate the domain generalization setting [28, 29]. Data augmentation is also a popular tool used for improving domain generalization. It ranges from introducing various types of augmentations to simulate unseen test domain conditions (E.g. style transfer [50, 52]) to self-supervised learning involving matching the representations of an image with different augmentations (E.g. [1, 7]). Finally, different ways of regularizing models (implicit and explicit) have also been developed with the goal of encouraging domain-invariant feature learning [38, 47, 46]. For instance, invariant risk minimization [2] propose a regularization such that the classifier is optimal in all the environments. Representation Self-Challenging [21] propose to suppress the dominant features that get activated on the training data, which forces the network to use other features that correlate with labels. Risk extrapolation [26] propose a regularization that minimizes the variance between domain-wise loss, in the hope that it is representative of the variance including unseen test domains. See [51] for a survey on DG methods.

Our investigation in this work is complementary to all these domain generalization methods. Additionally, one of our main focus is to also study and improve performance instability on out-domain data during training, which results in more reliable model selection. This aspect has not received much attention.

## 6 Conclusion

We investigated a hyperparameter-free and efficient protocol for model averaging in the ERM framework, and showed that it provides a significant boost to out-domain performance compared to un-averaged models. Building on this observation, we showed that an ensemble of moving average models performs better compared to an ensemble of un-averaged models. Importantly, we showed that in both cases, model averaging significantly improves the rank correlation between in-domain validation accuracy and out-domain test accuracy, which is crucial for reliable model selection using in-domain validation data. We experimented with three pre-trained models with increasing pre-training dataset and model size, and found that EoA provides a proportionally larger gain compared to the corresponding ERM baseline, and lies in the range of $4\% - 6\%$. Finally, we explain the performance boost of EoA by adapting the Bias-Variance trade-off perspective to the domain generalization setting. Further discussions along with limitations of our work are provided in Appendix E.

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
