Table 6: Hyper-parameter search space for all experiments.

| Hyper-parameter | Default value | Random distribution | |
|---|---|---|---|
| | | [18] | Ours |
| Learning rate | $5e-5$ | $10^{\text{Uniform}(-5,-3.5)}$ | $5e-5$ |
| Batch size | 32 | $2^{\text{Uniform}(3,5.5)}$ | 32 |
| ResNet dropout | 0 | RandomChoice([0, 0.1, 0.5]) | RandomChoice([0, 0.1, 0.5]) |
| Weight decay | 0 | $10^{\text{Uniform}(-6,-2)}$ | $10^{\text{Uniform}(-6,-4)}$ |

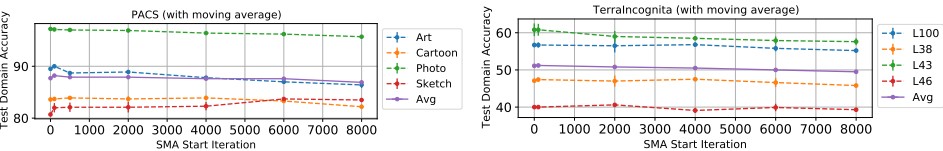

Figure 5: The impact of iteration $t_0$ at which we start simple moving averaging as described in Eq. 1, on the domain generalization performance for PACS and TerraIncognita datasets. The dominant pattern across all the experiments suggests that starting averaging earlier yields a stronger boost in performance.

# Appendix

# A    Broader Impact

Our work aims at improving out-domain performance of models. Thus it is naturally geared towards mitigating the effects of training dataset biases on the hypothesis learned by the model, which we believe has a positive societal impact.

# B    Training and Evaluation Protocols for DomainBed Benchmarking

We use the training protocol described in [18] with minor changes: we use a smaller hyper-parameter search space (shown in Table 6) and smaller number of random trials for computational reasons, and train on DomainNet dataset for $15,000$ iterations instead of $5,000$ similar to [8], because its training loss is quite high. For a dataset with $D$ domains, we run a total of $6D$ random trials. This results in 6 experiments per domain, in which this domain is used as the test set, while the remaining domains are used as training/validation set (randomly split). This is also the reason why we use a smaller hyper-parameter search space, because otherwise the search space would be under-sampled. For moving average models, the iteration $t_0$ at which averaging is started (Eq. 1) is set to be $100$ in all experiments unless specified otherwise. For ensembles (both with and without averaged models), the 6 models corresponding to the 6 experiments per domain, in which this domain is used as the test set (as described above), are used for ensembling as described in section 3.

All models are trained using the ERM objective and optimized using the Adam optimizer [25]. We use ResNet-50 [19] pre-trained on Imagenet as our initialization for training in all the experiments. In the final benchmarking experiment, we also use ResNeXt-50 32x4d, that is trained using semi-weakly supervised objective on IG-1B targeted (containing 1 billion weakly labeled images) and ImageNet labeled data [48]. This model was downloaded from Pytorch hub. For all models, the batch normalization [22] statistics are kept frozen throughout training and inference. Validation accuracy is calculated every 300 iterations for all datasets except DomainNet where it is calculated every 1000 iterations. Unless specified otherwise, we use the said protocol in all the experiments. For model selection, we use the *training-domain validation set* protocol in [18] with $80\% - 20\%$ training-validation split, and the average out-domain test performance is reported across all runs for each domain.

All experiments were performed on Google Could Platform (GCP) using 24 NVIDIA A100 GPUs.

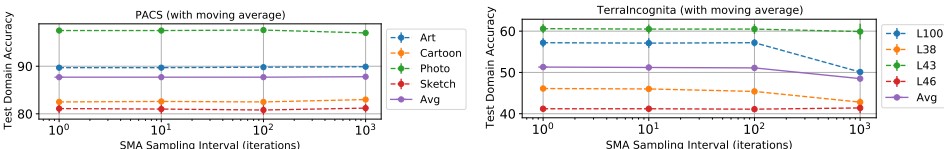

Figure 6: The impact of the frequency (number of iterations), at which model states are sampled for computing the simple moving average (SMA), on the domain generalization performance for PACS and TerraIncognita datasets. Broadly, we find that the frequency of sampling does not have a major influence on performance unless the sampling interval is too large: performance drops significantly on TerraIncognita only when the frequency is set to sampling every 1000 iterations.

# C  Additional Ablation Analysis of Our Model Averaging Protocol (Eq. 1)

## C.1  Dataset and Experiment Details for SMA Ablation Analysis in Section 2.3, C.2, and C.3

**Experimentation Details**: We use the training protocol described in [18] with minor changes: we use a smaller hyper-parameter search space for feasibility, and train on DomainNet dataset for $15,000$ iterations instead of $5,000$ similar to [8], because its training loss is quite high. Unless specified otherwise, we use the said protocol in all the experiments. For model selection, we use the *training-domain validation set* protocol in [18], where the average out-domain test performance is reported across all runs. For more details, see section B in the appendix.

**Dataset Details**: We use a subset of the DomainBed benchmark: PACS dataset (4 domains, 7 classes, and 9,991 images), TerraIncognita dataset (4 domains, 10 classes, and 24,788 images), VLCS dataset (4 domains, 5 classes, and 10,729 images), OfficeHome dataset (4 domains, 65 classes, and 15,588 images), and DomainNet dataset (6 domains, 345 classes, and 586,575 images).

## C.2  Start Iteration

We investigate how domain generalization performance is impacted by the choice of iteration $t_0$ when we start model averaging in Eq. 1. In this section, we simply refer to it as start iteration, which should not be confused with the start of the training process. For experiments, we use the PACS and TerraIncognita datasets. To investigate a wide range of start and end iterations, for all experiments in this section, we train models for $10,000$ iterations.

We consider starting model averaging from iterations in $\{0, 100, 500, 2000, 4000, 6000, 8000\}$. We plot the performance in Figure 5 for each value. We find that the test performance, averaged across all the domains for both datasets, decreases if we start model averaging later during the training, and choosing $t_0$ close to initialization yields the best performance. We believe using the initialization state in model averaging causes a slight dip in performance because loss is initially high. Based on these experiments, instead of tuning start iteration as a hyper-parameter, *we arbitrarily choose 100 as the start iteration for the remaining experiments in this paper*. This choice of starting averaging later during training is called *tail averaging*, and the theoretical motivation for this choice are discussed in more detail in section 5.1.

## C.3  Averaging Frequency

When performing simple model averaging described in Eq. 1, instead of averaging iterates from every iteration, we can alternatively sample iterates at a larger interval. We study the impact of averaging frequency on out-domain test performance. Once again, we use the PACS and TerraIncognita datasets. We train models for $10,000$ iterations, and sample iterates at intervals in $\{10^0, 10^1, 10^2, 10^3\}$ iterations. Test accuracy is once again computed using the protocol of [18] for each case. The performance as a function of the iterate sampling interval used in SMA is shown in Figure 6. Broadly, we find that the frequency of sampling does not have a major impact on performance unless the sampling interval is too large, which happens in the case of TerraIncognita, where performance drops significantly when the sampling interval is set to 1000.

### C.4 Rank Correlation

Rank correlation metrics aim to quantify the degree to which an increase in one random variable's value is consistent with an increase in the other random variable's value. Therefore, instead of Pearson's correlation, they are better suited for studying the relationship between the in-domain and out-domain performance for the purpose of model selection because we select the best model during an optimization based on ranking the validation performance (early stopping). We consider Spearman correlation in our experiments. Its value vary between $-1$ and $+1$, where $-1$ implies the ranking of the two random variables are exactly the reverse of each other, and $+1$ implies the ranking of the two random variables are exactly the same as each other. A value of $0$ implies there is no relationship between the two variables.

### C.5 Instability Reduction: Qualitative Analysis

Here we try to qualitatively study the robustness of model selection using in-domain validation set, on out-domain performance. To do so, consider the ideal scenario where the in-domain validation performance correlates well with the out-domain performance. In this case, training longer should not be a problem in general, because if the model starts overfitting beyond a certain point, model selection can take care of it. In such a situation, we would expect the out-domain performance to either improve with longer training, or remain stable.

We use TerraIncognita dataset for this experiment. We consider training duration to be $1,000$ to $10,000$ iterations, at intervals of $1,000$. We plot the performance in Figure 7 for online model (left) and moving average model (right). We find that the performance of moving average models is more stable compared to online models, suggesting that model selection is more reliable when using moving average models. Figure 12 in the appendix shows the training loss, in-domain validation accuracy and out-domain test accuracy for all the runs used in the above experiment. It shows that the out-domain test performance is unstable during optimization without model averaging, which causes problem for model selection using the in-domain validation performance, as is evident in the above experiment.

### C.6 Cross-run rank correlation

In addition to the experiments in 2.3.1, there is another way in which it makes sense to study the rank correlation between validation and test performance. Suppose we set one of the domains of PACS as our test domain, and the remaining as training/validation data, and perform multiple independent runs with different seeds/hyper-parameter values. At each iteration during training, we can gather the tuple (validation, test) accuracy for each of these runs, and then study the rank correlation between them. The utility of this perspective is to assess the reliability of model selection in terms of selecting a single model across multiple independently trained models, based on their validation performance. We study this rank correlation for PACS and TerraIncognita datasets. The results are shown in Figure 8 in the appendix. We find that the cross-run rank correlations are poor (not consistently close to 1) for both online model (without averaging) and moving average model. This implies that in-domain validation performance based model selection is not a reliable approach for selecting a model from a pool of multiple independently trained models.

## D  Why Does Ensembling Improve Performance?

We describe the histogram experiment presented in section 3.2. For this experiment, we use the TerraIncognita dataset with L46 as the test domain. We use one of the runs with model averaging from the experiments done in section 4.1. We begin by re-writing the Taylor's expansion more precisely for our model averaging protocol,

$$\frac{1}{T-t_0+1} \cdot \sum_{t=t_0}^{T} f(\mathbf{x};\theta_t)_k \approx f(\mathbf{x};\hat{\theta}_T)_k + 0.5 \cdot \frac{1}{T-t_0+1} \cdot \sum_{t=t_0}^{T} (\hat{\theta}_T - \theta_t)^T \frac{\partial^2 f(\mathbf{x};\hat{\theta}_T)_k}{\partial \hat{\theta}_T^2} (\hat{\theta}_T - \theta_t)$$

(3)

where $\hat{\theta}_T := \frac{1}{T-t_0+1} \cdot \sum_{t=t_0}^{T} \theta_t$. Notice the first order term has been omitted since it is zero. As an important detail, instead of computing the second order term in the above equation for all

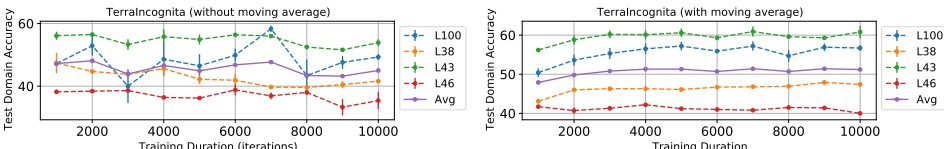

Figure 7: Qualitatively accessing the reliability of model selection while varying the training duration. For each training duration, the out-domain test accuracy is calculated using model selection over the in-domain validation data. Not using model averaging leads to unreliable model selection as evident in the instability of the out-domain performance. Model averaging is able to reduce this instability.

Table 7: Spearman correlation (closer to 1 is better) between within-run in-domain validation accuracy and out-domain test accuracy on the multiple datasets in the DomainBed benchmark for individual models (left) and ensembles (right). In most cases, using model averaging results in a significantly better rank correlation, which makes model selection more reliable.

| Table 8: Individual Models | | | Table 9: Ensembles | | |
|---|---|---|---|---|---|
| PACS | w/o avg | with avg | PACS | w/o avg | with avg |
| Art | $0.31 \pm 0.04$ | $\mathbf{0.62 \pm 0.04}$ | Art | 0.06 | **0.78** |
| Cartoon | $0.25 \pm 0.10$ | $\mathbf{0.52 \pm 0.03}$ | Cartoon | 0.33 | **0.81** |
| Photo | $\mathbf{0.09 \pm 0.07}$ | $-0.38 \pm 0.15$ | Photo | **-0.12** | -0.52 |
| Sketch | $0.24 \pm 0.06$ | $\mathbf{0.53 \pm 0.06}$ | Sketch | 0.43 | **0.70** |
| TerraIncognita | w/o avg | with avg | TerraIncognita | w/o avg | with avg |
| L100 | $0.21 \pm 0.07$ | $\mathbf{0.90 \pm 0.05}$ | L100 | 0.48 | 1 |
| L38 | $0.12 \pm 0.13$ | $\mathbf{0.83 \pm 0.05}$ | L38 | 0.17 | **0.95** |
| L43 | $0.30 \pm 0.06$ | $\mathbf{0.67 \pm 0.18}$ | L43 | **0.59** | 0.38 |
| L46 | $0.03 \pm 0.11$ | $\mathbf{0.52 \pm 0.14}$ | L46 | 0.08 | **0.61** |
| VLCS | w/o avg | with avg | VLCS | w/o avg | with avg |
| Caltech101 | $\mathbf{0.21 \pm 0.10}$ | $0.16 \pm 0.15$ | Caltech101 | 0.52 | **0.81** |
| LabelMe | $\mathbf{0.30 \pm 0.08}$ | $0.02 \pm 0.14$ | LabelMe | 0.05 | **0.38** |
| Sun09 | $0.27 \pm 0.12$ | $\mathbf{0.32 \pm 0.11}$ | Sun09 | 0.63 | **0.82** |
| VOC2007 | $0.17 \pm 0.11$ | $\mathbf{0.38 \pm 0.05}$ | VOC2007 | 0.55 | **0.65** |
| OfficeHome | w/o avg | with avg | OfficeHome | w/o avg | with avg |
| Art | $0.05 \pm 0.11$ | $\mathbf{0.80 \pm 0.04}$ | Art | 0.27 | **0.92** |
| Clipart | $0.33 \pm 0.04$ | $\mathbf{0.84 \pm 0.04}$ | Clipart | 0.66 | **0.95** |
| Product | $0.61 \pm 0.04$ | $\mathbf{0.80 \pm 0.04}$ | Product | 0.20 | **0.95** |
| RealWorld | $0.41 \pm 0.06$ | $\mathbf{0.74 \pm 0.04}$ | RealWorld | 0.09 | **0.78** |
| DomainNet | w/o avg | with avg | DomainNet | w/o avg | with avg |
| Clip | $0.96 \pm 0.01$ | $\mathbf{1 \pm 0}$ | Clip | 1 | 1 |
| Info | $0.80 \pm 0.05$ | $\mathbf{1 \pm 0}$ | Info | 0.88 | 1 |
| Paint | $0.87 \pm 0.02$ | $\mathbf{1 \pm 0}$ | Paint | 0.98 | 1 |
| Quick | $0.65 \pm 0.04$ | $\mathbf{1 \pm 0}$ | Quick | 0.95 | 1 |
| Real | $0.91 \pm 0.01$ | $\mathbf{1 \pm 0}$ | Real | 0.97 | 1 |
| Sketch | $0.82 \pm 0.04$ | $\mathbf{1 \pm 0}$ | Sketch | 0.97 | 1 |

integer valued $t$ between $t_0$ and $T$, we only use $\theta_t$ for $t = 300 * i$ for $i \in \{1, 2, \ldots, 16\}$ due to computational constraints. This should not affect our conclusion because as shown in section C.3, averaging frequency does not have a significant impact on performance. Finally, we record the logit values and the second order term for 1000 randomly selected samples $\mathbf{x}$ from the test domain data. Note that TerraIncognita has 10 classes and so we have the above equation for $k \in \{1, 2, \ldots, 10\}$. We plot the histogram of the logit values and the second order term corresponding to each class's output separately. All the plots are shown in figure 9. The conclusion is the same as described in the main text.

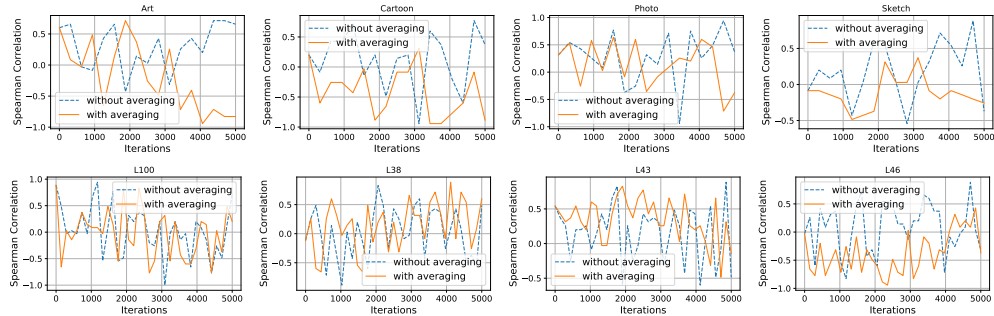

Figure 8: Spearman correlation between *cross-run* in-domain validation accuracy and out-domain test accuracy for PACS dataset (top) and TerraIncognita dataset (bottom). The cross-run rank correlations are poor (not consistently close to 1) for both online model (without avg) and moving average model. This implies that in-domain validation performance based model selection is not a reliable approach for selecting a single model from a pool of multiple independently trained models. See section 2.3.1 for details.

## E   Discussions and Limitations

**Domain Generalization Limitations**: The Bias-Variance trade-off analysis of EoA in section 3.2 shows that the expected loss of individual models constitutes both the bias and variance term, while that of ensembles is dominated by the bias term alone, and is thus strictly lower. Thus ensembles (either explicitly or implicitly by model averaging) is guaranteed to improve performance on OOD data compared to the corresponding individual unaveraged models. However, this strategy cannot go beyond getting rid of the variance term, and other strategies need to be used to reduce the bias term, which will further improve the OOD performance.

Additionally, our proposal does not make use of the environment ID of samples. A popular strategy to utilize this information is to increase some form of alignment between the latent representations of different domains, which has been shown to be one of the terms in the upper bound of the test domain generalization error [4]. While a number of prior work has proposed variants of this strategy (eg. CORAL, DANN, see section 5.2 for a discussion), [18] has showed that ERM with appropriate training-validation protocol performs at least as well as these methods. Further, [44] has recently argued supported by empirical evidence that domain alignment is neither necessary nor sufficient for domain generalization. Thus it remains an open question what other strategies can be used to utilize the environment ID to boost domain generalization

**Functional Diversity**: Model averaging mitigates instability within a run, which makes model selection more reliable. However, we note that the gap in performance between different runs still exists, though it is smaller on average compared with online models (see training evolution plots in appendix for reference). On another note, while our analysis in section 3.2 suggests that model averaging behaves as an ensemble, we believe that it does not offer as much of functional diversity as independently trained models. This is because if it did, model averaging should have had a much better performance compared with a traditional ensembles (since there are $T - t_0 + 1$ models in model averaging while only 6 models in the traditional ensemble in our experiments), but this is not the case (see their performance in table 10 for a comparison). This implies that there is still diversity among the independently trained models with model averaging. This is also inline with [14] which shows that ensembling methods such as model averaging and Monte Carlo dropout [15] do not provide diversity in function space as much as ensembles of independently trained models. Perhaps this is why EoA performs better compared to both individual moving average models and traditional ensembles, by better approximating the expected ensemble behavior.

**Scalability**: Following the protocol of [18], we used samples from all the training domains in each mini-batch update. However, in settings where the number of domains is very large, this approach can be prohibitive. As an alternative, we also performed preliminary experiments in which we stochastically picked one of the training domains at every iteration, and sampled a mini-batch from that domain to update parameters. We found that this protocol resulted in a similar performance as that achieved by the protocol used in our work.

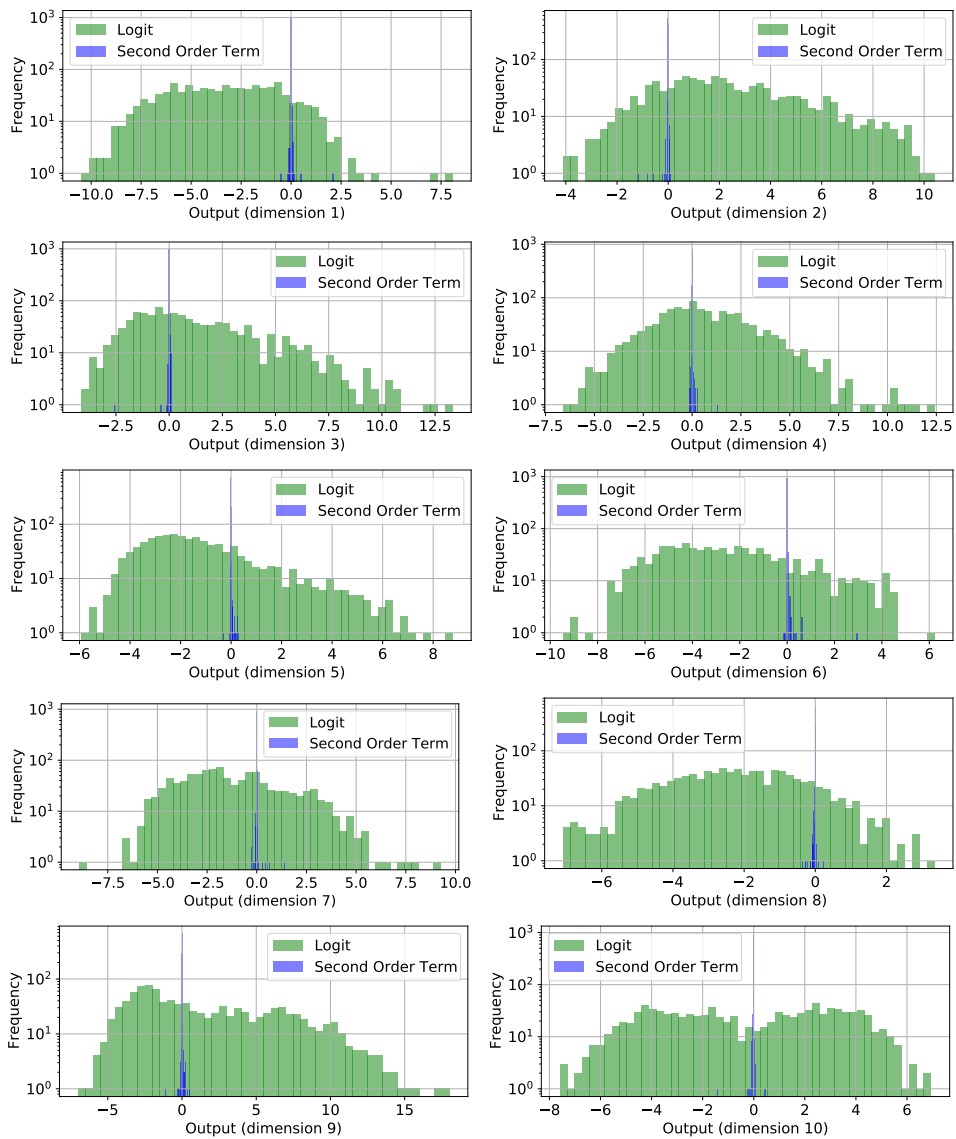

Figure 9: The scale of terms– moving average model's logit and the second order term in Eq. 3. The latter concentrates around 0, suggesting our model averaging protocol approximates ensembles.

**Computational Complexity**: The computational overhead due to SMA is practically negligible (compared to back-propagation) since it merely involves a running average estimate of the parameters. So its complexity is the same as that of training a vanilla supervised deep network. On the other hand, since EoA trains an ensemble of models, the complexity scales linearly with the number of models used in the ensemble compared with SMA on a single model, if these models are trained sequentially. Of course, these different models can be trained in parallel if resources are available, given each model is trained independently of one another, in which case the complexity of EoA remains the same as that of vanilla supervised training.

**Information leaking considerations**: We present many experiments where validation and test performances are studied. It is therefore natural to wonder if there was any information leak from the test set while performing this analysis. We note that in the model averaging protocol we investigated, there were two moving parts: iteration $t_0$ at which model averaging is started, and averaging frequency. We studied them in section C.2 and C.3 respectively. For both of them, we proposed to fix their values universally instead of tuning them on each dataset. Specifically, [24] propose tail averaging in which iterates from every iteration are used for computing the simple moving average. We found this proposal to work well empirically in our analysis, and therefore set averaging frequency to 1. For start

Table 10: Performance benchmarking on 5 datasets of the DomainBed benchmark using two different pre-trained models. SWAD is the previous SOTA. Note that ensembles do not have confidence interval because an ensemble uses all the models to make a prediction. Gray background shows our proposal. *Our runs* implies we ran experiments, but we did not propose it.

| Algorithm | PACS | VLCS | OfficeHome | TerraIncognita | DomainNet | Avg. |
|---|---|---|---|---|---|---|
| ResNet-50 (25M Parameters, pre-trained on ImageNet) | | | | | | |
| ERM (our runs) | $84.4 \pm 0.8$ | $77.1 \pm 0.5$ | $66.6 \pm 0.2$ | $48.3 \pm 0.2$ | $43.6 \pm 0.1$ | 64.0 |
| Ensemble (our runs) | 87.6 | 78.5 | 70.8 | 49.2 | **47.7** | 66.8 |
| ERM [18] | $85.7 \pm 0.5$ | $77.4 \pm 0.3$ | $67.5 \pm 0.5$ | $47.2 \pm 0.4$ | $41.2 \pm 0.2$ | 63.8 |
| IRM [2] | $84.4 \pm 1.1$ | $78.1 \pm 0.0$ | $66.6 \pm 1.0$ | $47.9 \pm 0.7$ | $35.7 \pm 1.9$ | 62.5 |
| Group DRO [38] | $84.1 \pm 0.4$ | $77.2 \pm 0.6$ | $66.9 \pm 0.3$ | $47.0 \pm 0.3$ | $33.7 \pm 0.2$ | 61.8 |
| Mixup [47, 46] | $84.3 \pm 0.5$ | $77.7 \pm 0.4$ | $69.0 \pm 0.1$ | $48.9 \pm 0.8$ | $39.6 \pm 0.1$ | 63.9 |
| MLDG [28] | $84.8 \pm 0.6$ | $77.1 \pm 0.4$ | $68.2 \pm 0.1$ | $46.1 \pm 0.8$ | $41.8 \pm 0.4$ | 63.6 |
| CORAL [41] | $86.0 \pm 0.2$ | $77.7 \pm 0.5$ | $68.6 \pm 0.4$ | $46.4 \pm 0.8$ | $41.8 \pm 0.2$ | 64.1 |
| MMD [30] | $85.0 \pm 0.2$ | $76.7 \pm 0.9$ | $67.7 \pm 0.1$ | $49.3 \pm 1.4$ | $39.4 \pm 0.8$ | 63.6 |
| DANN [16] | $84.6 \pm 1.1$ | $78.7 \pm 0.3$ | $65.4 \pm 0.6$ | $48.4 \pm 0.5$ | $38.4 \pm 0.0$ | 63.1 |
| C-DANN [31] | $82.8 \pm 1.5$ | $78.2 \pm 0.4$ | $65.6 \pm 0.5$ | $47.6 \pm 0.8$ | $38.9 \pm 0.1$ | 62.6 |
| Fish [39] | $85.5 \pm 0.3$ | $77.8 \pm 0.3$ | $68.6 \pm 0.4$ | $45.1 \pm 1.3$ | $42.7 \pm 0.2$ | 63.9 |
| Fishr [37] | $85.5 \pm 0.4$ | $77.8 \pm 0.1$ | $67.8 \pm 0.1$ | $47.4 \pm 1.6$ | $41.7 \pm 0.0$ | 65.7 |
| SWAD [8] | $88.1 \pm 0.4$ | $\mathbf{79.1 \pm 0.4}$ | $70.6 \pm 0.3$ | $50.0 \pm 0.4$ | $46.5 \pm 0.2$ | 66.9 |
| MIRO [9] | $85.4 \pm 0.4$ | $79.0 \pm 0.$ | $70.5 \pm 0.4$ | $50.4 \pm 1.1$ | $44.3 \pm 0.2$ | 65.9 |
| SMA (ours) | $87.5 \pm 0.2$ | $78.2 \pm 0.2$ | $70.6 \pm 0.1$ | $50.3 \pm 0.5$ | $46 \pm 0.1$ | 66.5 |
| EoA (ours) | **88.6** | **79.1** | **72.5** | **52.3** | 47.4 | **68.0** |
| ResNeXt-50 32x4d [48] (25M Parameters, Pre-trained 1B Images) | | | | | | |
| ERM (our runs) | $88.9 \pm 0.3$ | $79.0 \pm 0.1$ | $70.9 \pm 0.5$ | $51.4 \pm 1.2$ | $48.1 \pm 0.2$ | 67.7 |
| Ensemble (our runs) | 91.2 | 80.3 | 77.8 | 53.5 | 52.8 | 71.1 |
| SMA (ours) | $92.7 \pm 0.3$ | $79.7 \pm 0.3$ | $78.6 \pm 0.1$ | $53.3 \pm 0.1$ | $53.5 \pm 0.1$ | 71.6 |
| EoA (ours) | **93.2** | **80.4** | **80.2** | **55.2** | **54.6** | **72.7** |
| RegNetY-16GF [40] (81M Parameters, Pre-trained on 3.6B Images) | | | | | | |
| ERM (our runs) | $92.0 \pm 0.4$ | $78.6 \pm 0.6$ | $73.8 \pm 0.5$ | $55.6 \pm 0.9$ | $53.2 \pm 0.2$ | 70.6 |
| Ensemble (our runs) | 95.1 | 80.6 | 80.5 | 59.5 | 57.8 | 74.7 |
| ERM [9] | $89.6 \pm 0.4$ | $78.6 \pm 0.3$ | $71.9 \pm 0.6$ | $51.4 \pm 1.8$ | $48.5 \pm 0.6$ | 68.0 |
| SWAD [9] | $94.7 \pm 0.2$ | $79.7 \pm 0.2$ | $80.0 \pm 0.1$ | $57.9 \pm 0.7$ | $53.6 \pm 0.6$ | 73.2 |
| MIRO [9] | $\mathbf{97.4 \pm 0.2}$ | $79.9 \pm 0.6$ | $80.4 \pm 0.2$ | $58.9 \pm 1.3$ | $53.8 \pm 0.1$ | 74.1 |
| SMA (ours) | $95.5 \pm 0.0$ | $80.7 \pm 0.1$ | $82.0 \pm 0.0$ | $59.7 \pm 0.0$ | $60.0 \pm 0.0$ | 75.6 |
| EoA (ours) | 95.8 | **81.1** | **83.9** | **61.1** | **60.9** | **76.6** |

iteration $t_0$ on the other hand, [24] theoretically show that the initial bias term in the excess error upper bound decays exponentially with $t_0$. In line with this theory, our analysis showed that an iteration close to but not at initialization worked well. So we arbitrarily set $t_0 = 100$. Note that these are not their optimal values, but are rather arbitrary choices guided by our investigation and existing theory. Aside from these two objects (which we fix in all experiments except the aforementioned ablation), there are no hyper-parameters introduced by the averaging protocol or the ensemble of averages studied in this paper, and all other experiments are purely observational. Finally, we followed the protocol of [18] for training and evaluation.

**Smaller HP search space**: We use a smaller hyper-parameter search space compared with that in [18]. Nonetheless, we find that on average, our runs of the ERM baseline performance (without model averaging) yield 64% test accuracy on average compared with 63.8% reported in [18] on the 5 datasets we used. We also note that model averaging and ensemble of averages, that we study in our work, are not competing with ERM baseline, in the sense that these techniques essentially rely on the quality of the baseline model to further boost performance. Therefore any boost in the ERM baseline performance is likely to improve the model averaging and EoA performance. This is also evident in our benchmarking experiments in Table 4, where using ResNeXt-50 32x4d pre-trained on a larger dataset [48] has a better ERM baseline performance compared to ResNet-50 pre-trained on ImageNet. This results in a further boost of 3.9% and 5% test accuracy on average when using model averaging and EoA respectively.

Table 11: Out-domain accuracy for PACS dataset.

| Algorithm | A | C | P | S | Avg. |
|---|---|---|---|---|---|
| ResNet-50 | | | | | |
| ERM | $86.4 \pm 1.0$ | $80.4 \pm 0.6$ | $94.8 \pm 0.1$ | $76.2 \pm 1.7$ | 84.4 |
| Ensemble | 88.3 | **83.6** | 96.5 | 81.9 | 87.6 |
| SMA | $89.1 \pm 0.1$ | $82.6 \pm 0.2$ | $97.6 \pm 0.0$ | $80.5 \pm 0.9$ | 87.5 |
| Ensemble of Averages (EoA) | **90.5** | 83.4 | **98.0** | **82.5** | **88.6** |
| ResNeXt-50 32x4d [48] | | | | | |
| ERM | $84.7 \pm 1.6$ | $87.6 \pm 0.1$ | $97.6 \pm 0.4$ | $85.7 \pm 0.1$ | 88.9 |
| Ensemble | 90.2 | 89.2 | 98.1 | 87.2 | 91.2 |
| SMA | $92.6 \pm 0.3$ | $90.9 \pm 0.8$ | $99.1 \pm 0.3$ | $88.3 \pm 0.5$ | 92.7 |
| Ensemble of Averages (EoA) | **93.1** | **91.8** | **99.2** | **88.9** | **93.2** |
| RegNetY-16GF [40] | | | | | |
| ERM | $90.2 \pm 0.6$ | $92.6 \pm 0.8$ | $97.6 \pm 0.1$ | $87.8 \pm 2$ | 92 |
| Ensemble | 93.75 | 95.35 | 98.02 | 93.38 | 95.1 |
| SMA | $93.8 \pm 0.3$ | $95.8 \pm 0.2$ | $99.2 \pm 0.2$ | **$93.4 \pm 0.2$** | 95.5 |
| Ensemble of Averages (EoA) | **94.09** | **96.33** | **99.52** | 93.31 | **95.8** |

Table 12: Out-domain accuracy for VLCS dataset.

| Algorithm | C | L | S | V | Avg. |
|---|---|---|---|---|---|
| ResNet-50 | | | | | |
| ERM | $98.5 \pm 0.5$ | $62.4 \pm 1.4$ | $72.1 \pm 0.0$ | $75.4 \pm 0.1$ | 77.1 |
| Ensemble | 98.7 | **64.5** | 72.1 | **78.9** | 78.5 |
| SMA | $99.0 \pm 0.2$ | $63.0 \pm 0.2$ | $74.5 \pm 0.3$ | $76.4 \pm 1.1$ | 78.2 |
| Ensemble of Averages (EoA) | **99.1** | 63.1 | **75.9** | 78.3 | **79.1** |
| ResNeXt-50 32x4d [48] | | | | | |
| ERM | $97.0 \pm 0.4$ | $67.8 \pm 0.7$ | $75.7 \pm 0.2$ | $75.5 \pm 0.6$ | 79.0 |
| Ensemble | 98.4 | **66.1** | 76.4 | 80.5 | 80.3 |
| SMA | **$98.8 \pm 0.2$** | $63.3 \pm 0.6$ | $77.7 \pm 0.2$ | $79.2 \pm 0.8$ | 79.7 |
| Ensemble of Averages (EoA) | 98.7 | 64.1 | **78.2** | **80.6** | **80.4** |
| RegNetY-16GF [40] | | | | | |
| ERM | $96.0 \pm 0.4$ | $66.0 \pm 0.8$ | $76.1 \pm 0.7$ | $76.2 \pm 2.1$ | 78.6 |
| Ensemble | 97.88 | **67.28** | 78.46 | 78.55 | 80.6 |
| SMA | $98.1 \pm 0.2$ | $65.7 \pm 1.1$ | $79.2 \pm 1.1$ | $79.6 \pm 0.2$ | 80.7 |
| Ensemble of Averages (EoA) | **98.23** | 66.00 | **79.49** | **80.63** | **81.1** |

Table 13: Out-domain accuracy for OfficeHome dataset.

| Algorithm | A | C | P | R | Avg. |
|---|---|---|---|---|---|
| ResNet-50 | | | | | |
| ERM | $60.5 \pm 0.7$ | $54.5 \pm 0.8$ | $74.7 \pm 0.8$ | $76.6 \pm 0.2$ | 66.6 |
| Ensemble | 65.6 | 58.5 | 78.7 | 80.5 | 70.8 |
| SMA | $66.7 \pm 0.5$ | $57.1 \pm 0.1$ | $78.6 \pm 0.1$ | $80.0 \pm 0$ | 70.6 |
| Ensemble of Averages (EoA) | **69.1** | **59.8** | **79.5** | **81.5** | **72.5** |
| ResNeXt-50 32x4d [48] | | | | | |
| ERM | $64.7 \pm 1.0$ | $60.6 \pm 0.3$ | $77.1 \pm 0.4$ | $81.3 \pm 0.2$ | 70.9 |
| Ensemble | 74.1 | 67.3 | 83.9 | 86.0 | 77.8 |
| SMA | $76.7 \pm 0.4$ | $67.8 \pm 0.0$ | $84.0 \pm 0.1$ | $85.8 \pm 0.1$ | 78.6 |
| Ensemble of Averages (EoA) | **79.0** | **70.0** | **85.2** | **86.5** | **80.2** |
| RegNetY-16GF [40] | | | | | |
| ERM | $70.7 \pm 1.3$ | $60.0 \pm 0.5$ | $82.4 \pm 0.5$ | $82.1 \pm 0.4$ | 73.8 |
| Ensemble | 79.44 | 68.68 | 86.28 | 87.63 | 80.5 |
| SMA | $81.1 \pm 0.4$ | $72.3 \pm 0.6$ | $86.6 \pm 0.1$ | $88.2 \pm 0.1$ | 82.0 |
| Ensemble of Averages (EoA) | **83.89** | **73.95** | **88.22** | **89.37** | **83.9** |

Table 14: Out-domain accuracy for TerraIncognita dataset.

| Algorithm | L100 | L38 | L43 | L46 | Avg. |
|---|---|---|---|---|---|
| ResNet-50 | | | | | |
| ERM | $52.9 \pm 3.3$ | $43.3 \pm 1.7$ | $56.9 \pm 0.4$ | $40.2 \pm 2.1$ | 48.3 |
| Ensemble | 53.0 | 42.6 | 60.5 | 40.8 | 49.2 |
| SMA | $54.9 \pm 0.4$ | $45.5 \pm 0.6$ | $60.1 \pm 1.5$ | $40.5 \pm 0.4$ | 50.3 |
| Ensemble of Averages (EoA) | **57.8** | **46.5** | **61.3** | **43.5** | **52.3** |
| ResNeXt-50 32x4d [48] | | | | | |
| ERM | $64.0 \pm 0.0$ | $44.7 \pm 3.2$ | $56.1 \pm 3.0$ | $40.9 \pm 1.4$ | 51.4 |
| Ensemble | **65.7** | 43.1 | 62.6 | 42.6 | 53.5 |
| SMA | $60.1 \pm 1.0$ | $\mathbf{47.3 \pm 1.4}$ | $61.0 \pm 1.7$ | $44.9 \pm 0.8$ | 53.3 |
| Ensemble of Averages (EoA) | 63.5 | 46.0 | **64.3** | **46.9** | **55.2** |
| RegNetY-16GF [40] | | | | | |
| ERM | $67.1 \pm 2.8$ | $46.3 \pm 2.9$ | $61.4 \pm 0$ | $47.5 \pm 1.9$ | 55.6 |
| Ensemble | 71.67 | 50.76 | 64.11 | **51.45** | 59.5 |
| SMA | $72.4 \pm 0.0$ | $52.0 \pm 0.5$ | $66.8 \pm 0.4$ | $47.4 \pm 0.2$ | 59.7 |
| Ensemble of Averages (EoA) | **73.80** | **52.60** | **68.19** | 49.75 | **61.1** |

Table 15: Out-domain accuracy for Domainet dataset.

| Algorithm | clip | info | paint | quick | real | sketch | Avg. |
|---|---|---|---|---|---|---|---|
| ResNet-50 | | | | | | | |
| ERM | $63.4 \pm 0.2$ | $20.6 \pm 0.1$ | $50.0 \pm 0.1$ | $13.8 \pm 0.4$ | $62.1 \pm 0.2$ | $51.9 \pm 0.3$ | 43.6 |
| Ensemble | **68.3** | 23.1 | 54.5 | 16.3 | **66.9** | 57.0 | **47.7** |
| SMA | $64.4 \pm 0.3$ | $22.4 \pm 0.2$ | $53.4 \pm 0.3$ | $15.4 \pm 0.1$ | $64.7 \pm 0.2$ | $55.5 \pm 0.1$ | 46.0 |
| Ensemble of Averages (EoA) | 65.9 | **23.4** | **55.3** | **16.5** | 66.4 | **57.1** | 47.4 |
| ResNeXt-50 32x4d [48] | | | | | | | |
| ERM | $68.8 \pm 0.1$ | $25.5 \pm 0.1$ | $55.9 \pm 0.3$ | $14.7 \pm 0.7$ | $65.8 \pm 0.4$ | $58.0 \pm 0.4$ | 48.1 |
| Ensemble | 74.3 | 28.7 | 61.1 | 17.0 | 71.9 | 63.5 | 52.8 |
| SMA | $73.7 \pm 0.1$ | $29.9 \pm 0.0$ | $62.8 \pm 0.1$ | $18.1 \pm 0.1$ | $73.0 \pm 0.2$ | $63.6 \pm 0.4$ | 53.5 |
| Ensemble of Averages (EoA) | **74.6** | **31.3** | **63.7** | **19.3** | **73.6** | **65.1** | **54.6** |
| RegNetY-16GF [40] | | | | | | | |
| ERM | $74.7 \pm 0.1$ | $34.8 \pm 0.9$ | $60.3 \pm 0$ | $15.2 \pm 0.6$ | $71.1 \pm 0.3$ | $62.1 \pm 0.1$ | 53.1 |
| Ensemble | 79.12 | 38.71 | 65.89 | 18.13 | 76.43 | 68.40 | 57.8 |
| SMA | $78.8 \pm 0.1$ | $43.0 \pm 0.0$ | $68.6 \pm 0.0$ | $21.2 \pm 0.0$ | $78.5 \pm 0.1$ | $69.8 \pm 0.1$ | 60 |
| Ensemble of Averages (EoA) | **79.63** | **44.02** | **69.57** | **22.46** | **77.95** | **71.69** | **60.9** |

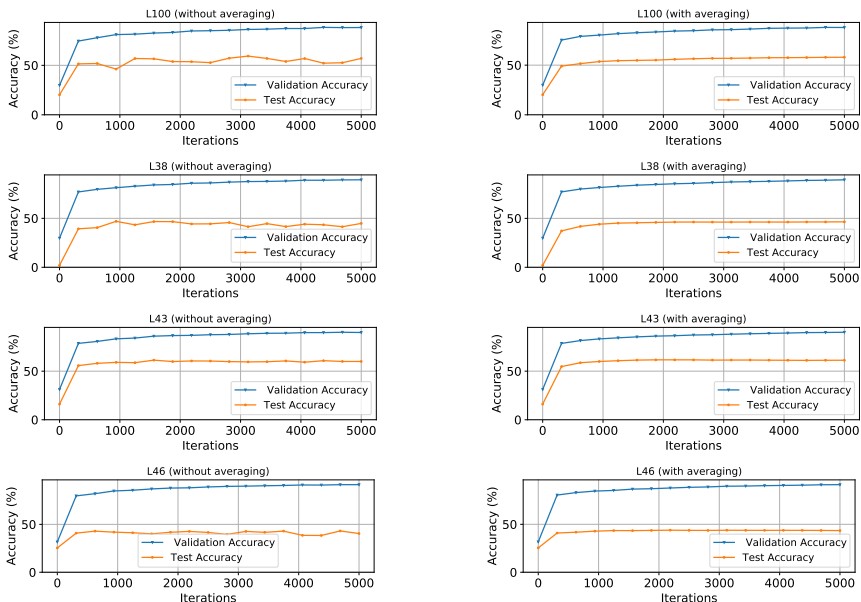

Figure 10: Ensemble of moving average (EOA) models (right) has better out-domain test performance *stability* compared with ensemble of online models (left), w.r.t. in-domain validation accuracy. **Details**: The plots are for the TerraIncognita dataset with the domain name in title used as the test domain, and others as training/validation domain, and ResNet-50. Each ensemble has 6 different models from independent runs with a different random seed and training/validation split.

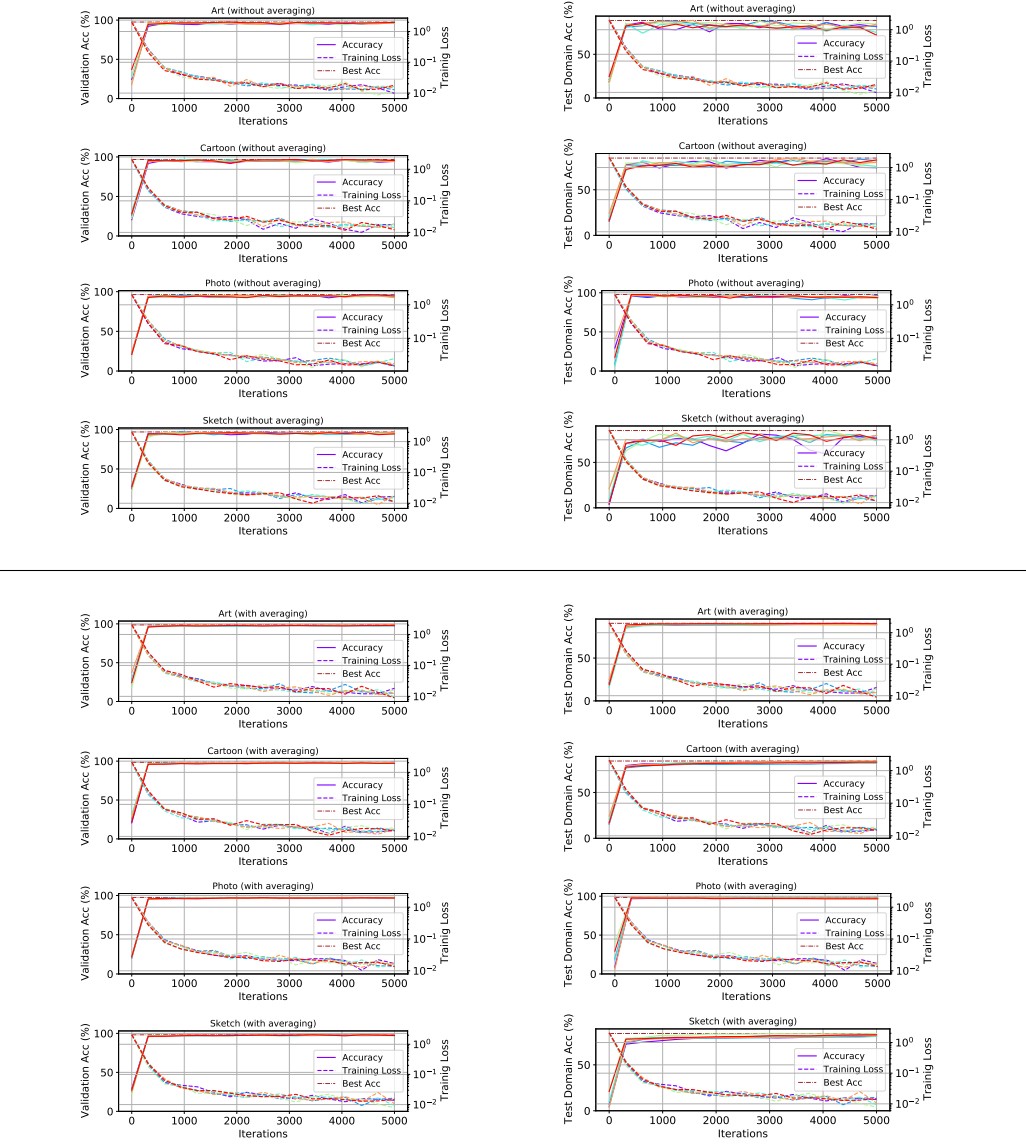

Figure 11: Evolution of training loss, in-domain validation accuracy and out-domain test accuracy for ResNet-50 (pre-trained on ImageNet) trained on PACS without model averaging (top 4 rows) and with model averaging (bottom 4 rows) for $5,000$ iterations with the domain mentioned in the title used as test domain and remain domains as training/validation data. Each color represents a different run with randomly chosen seed, hyper-parameters and training-validation split following [18].

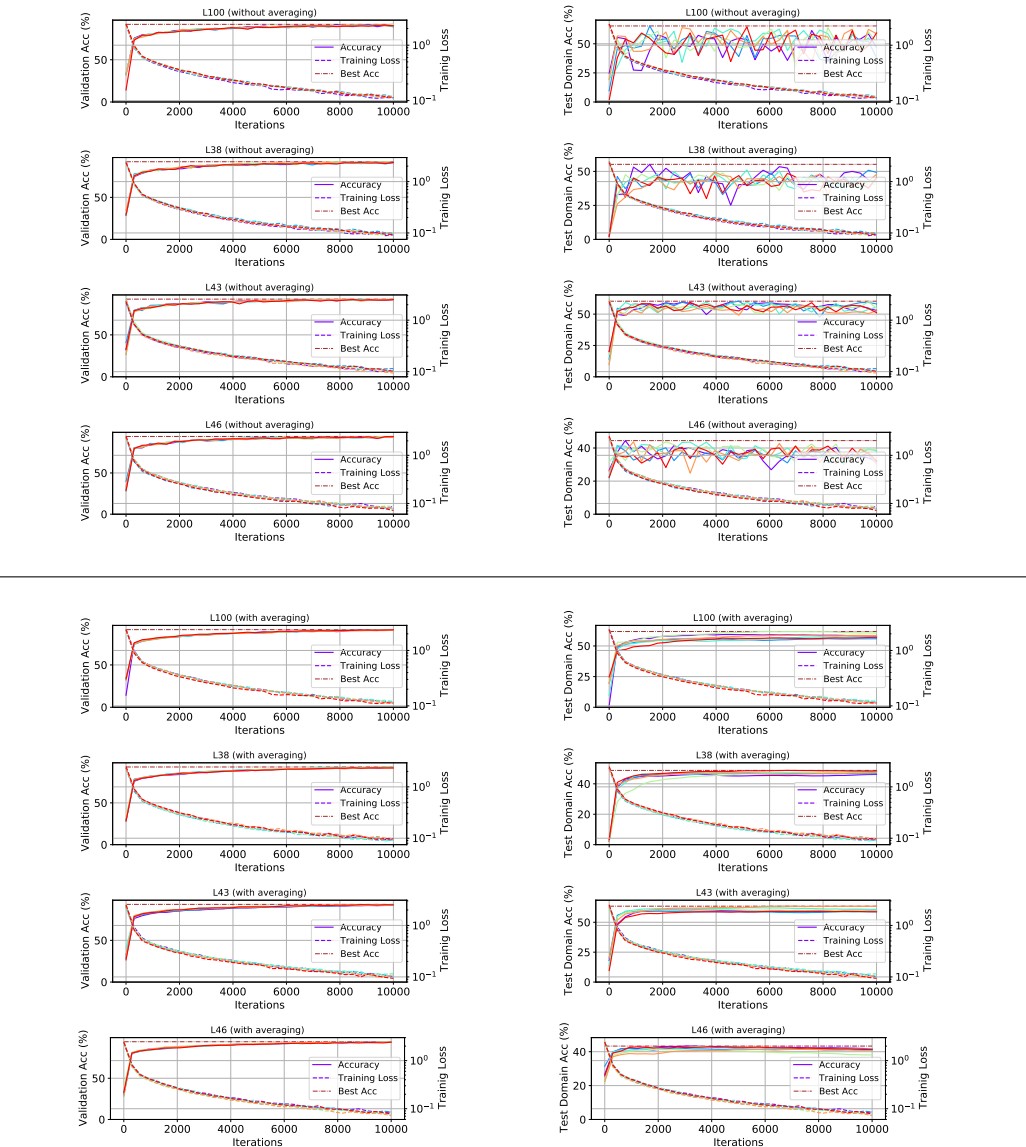

Figure 12: Evolution of training loss, in-domain validation accuracy and out-domain test accuracy for ResNet-50 (pre-trained on ImageNet) trained on TerraIncognita without model averaging (top 4 rows) and with model averaging (bottom 4 rows) for $10,000$ iterations with the domain mentioned in the title used as test domain and remain domains as training/validation data. Each color represents a different run with randomly chosen seed, hyper-parameters and training-validation split following [18]. **Gist**: Out-domain test performance is unstable without model averaging, which causes problem for model selection using in-domain validation performance. Model averaging is able to mitigate this instability.