# OpenReview forum: "Ensemble of Averages: Improving Model Selection and Boosting Performance in Domain Generalization"
_NeurIPS.cc/2022/Conference — NeurIPS 2022 Accept_

### Official Review · Reviewer_vuPK · 2022-06-19

**Rating:** 6
**Confidence:** 4
**Soundness:** 2 fair
**Presentation:** 1 poor
**Contribution:** 3 good

**Summary:**

This paper proposes an approach to improve the test accuracy of the model when evaluated on a test dataset from a different domain. The methods used for this purpose include the moving average of model weights and the ensemble of moving averaged models. Empirical experiments are conducted to show the effectiveness of the proposed method.

**Questions:**

* Ablation study 1) on the impact of the start iteration t0. The authors claim that “starting averaging close to initialization results in improved out-domain performance XXX”. The reviewer is not convinced by this statement based on the empirical results. It is true that  there is an increase of test accuracy at iteration 100, but this does not imply the statement the authors have. If the statement is correct, does the method gives the best performance at iteration 10 or even 1? Finer grids and more solid experiment results are needed for the statement.
* In the analysis on the ensemble in Section 3.2, in the Taylor expansion term, it is shown that the output of EoA is roughly the same as the model evaluated at $\hat \theta_T,$ which is the average of the model parameters over time. Can you explain why $f(\hat \theta_T)$ leads to a better performance?
* The experiment in Appendix C.6 on cross-run rank correlation claims that “in-domain validation performance based model selection is not a reliable approach for selecting a model from a pool of multiple independently trained models.” The statement seems interesting. If the statement is true, then how do you choose your model in practice since conceptually we have a pool of models trained with different learning rates, batch size, etc. In this case, how do you perform model selection?
* In Table 8 for the spearman correlation,  in for example Photo dataset, with average leads to negative correlation. You have claimed that closer to 1 is better. However, -0.38 implies a stronger correlation between two outputs then 0.09. Can we claim that with average still make the early stopping reliable given that they are negatively correlated?
* The paper needs to be better proofread:
	* L76, \cdot not \ldot
	* L87 reference needed for Polyak-Ruppert averaging
	* L201, should not include “not”?

**Limitations:**

I really like that the authors have a full section discussing the limitations of their work. The limitations and questions of the paper the reviewer had is given in Questions section.


**Strengths And Weaknesses:**

* Originality: The techniques used in the paper are not new and are well studied in other areas. The use of these techniques in domain generalization is however new. Related work is cited and discussed.
* Quality: The methods are used appropriate, the work is complete and I really like the fact that the authors are not overselling their work. However, I am not fully convinced by some of the arguments in the paper based on the experiment results. See Questions for details.
* Clarity: The presentation needs some improvement.
	* There are a lot of cross references in the paper that makes the paper a bit difficult to follow. For example, in section 2.3 of the ablation study, the authors refer the readers to Appendix C for the experiment result where Appendix C refer the readers to Appendix B for experiment details. The reviewers find a lot of such references in the paper.
	* The paper cited a lot of reference for experimental details (see for example L150 for the protocol), it is therefore unclear to judge purely from this paper whether an expert is able to reproduce the paper. The code for the paper is submitted in the supplementary material, which is helpful for reproducing purposes.
	* The figures in the paper need to be presented in a different way. It is often the case that for the test accuracy the y-axis have only two ticks 0 and 50 and it is very hard to tell the information from the figure.
* Significance: Based on the experiment result, the proposed ensemble method can beat the SOTA while the SMA has comparable performance with the SOTA. While the performance of SMA is not as good as the SOTA, the authors claim that the proposed method has computational efficiency. The results, while not earthshaking, may be of practical use in real applications.

---

> ### Author Response · Authors · 2022-07-30
> **Thank you for your comments. Please find our response below.**
>
> **The authors claim that “starting averaging close to initialization results in improved out-domain performance…If the statement is correct, does the method gives the best performance at iteration 10 or even 1?**:
> The phrase "*close* to initialization" is a qualitative statement. We are not claiming that the performance gets better the closer to the initialization we start model averaging, which you are implying. We hope this clarifies the confusion.
>
> **in Section 3.2, in the Taylor expansion term, it is shown that the output of EoA is roughly the same as the model evaluated at $\\theta\_T$...**:
> There seems to be a confusion here. We do not say that $\\hat{\\theta}\_T$ is roughly the same as EoA. We show that under the second order Taylor's expansion, $f(\\hat{\\theta}\_T)$ (i.e., the SMA model) approximates an ensemble of models along the optimization trajectory. But as discussed in the Limitations section in Appendix E, SMA does not have as much functional diversity compared to an ensemble of independently trained models. However, to boost the out-domain generalization performance based on our bias-variance decomposition analysis in section 3.2, we want to achieve the expected ensemble model behavior (i.e. containing many models), which is practically infeasible. Therefore our approach was to propose EoA as a middle ground between SMA and the expected ensemble model, which has better diversity compared to a traditional ensemble of the same finite size, but is computationally cheaper by taking advantage of SMA. We hope this is clearer.
>
> **The experiment in Appendix C.6 on cross-run rank correlation…how do you perform model selection?**:
> We assume that you are referring to the pool of models in the ensemble in our experiments. For ensembles, we do not choose any one specific model in our experiments. We use all the independently trained models (that use different hyper-parameters). The choice of the state of each model along its optimization trajectory is done using the SMA model on the in-domain validation set.
>
> If you meant how to choose a single model out of a pool of models in general for out-domain inference, to our knowledge, this is still an open question in the domain generalization setting.
>
> **In Table 8 for the spearman correlation for Photos dataset in PACS…negative correlation -0.38…**:
> To explain the negative correlation on PACS Photo domain, we note that test performance on the Photo domain converges to $\sim 99\%$ soon after training begins (because the photo domain is close to the ImageNet dataset, which is the dataset on which the model is pre-trained on). Therefore the correlation between validation and test accuracy is largely noise. This can be verified from Fig 11, second from the last row (appendix page 24). It can be seen that both the test accuracy converges around ~500 iterations. This is not the case for the other domains in PACS, where the rank correlations are positive and stronger in all cases.
>
> **L87 reference needed for Polyak-Ruppert averaging**:
> We will add a citation on Line 87. But for reference, the citation is provided in the related work section on Line 283.
>
> **Line 201, should not include “not”?**:
> The use of the word "not" is intentional and correct. We are stressing the fact that the first and second order derivatives in the Taylor's expansion in section 3.2 are not derivatives with respect to the loss function, but rather derivatives with respect to the model output.

---

> > ### Comment · Reviewer_vuPK · 2022-08-08
> > **Response**
> >
> > Thank you for your clarification and my belated response. I have two more questions.
> >
> > 1. Just out of curiosity, could there be any theoretical justification when to start averaging (i.e. the choice of $t_0$)?
> > 2. The experiment on Photos dataset. Thank you for your explanation, I think there is still some misunderstanding of spearmean's correlation. In my original review, I meant that the spearmean's correlation of -0.38 is **stronger** than 0.09, but the two terms are just negatively related. Therefore, this number supports your claim that with averaging improves the rank correlation between in-domain validation performance and out-domain test performance.  I do not agree with your statement that "Spearman correlation (closer to 1 is better) ". Following this point, it is interesting to study why on such a dataset with slight domain shift, the in-domain validation performance and out-domain test performance are negatively correlated.

---

> > > ### Author Response · Authors · 2022-08-08
> > > **Response**
> > >
> > > Thank you for your questions.
> > >
> > > 1. Intuitively, the value of $t\_0$ acts as a trade-off between the bias and the variance term in the bias-variance decomposition discussed in our work. Specifically, we find that the expected loss on the out-domain data can be factorized into a bias and a variance term. By using an ensemble, we can mitigate the effect of the variance term. Therefore, to make the expected out-domain loss smaller, we want to make the bias term as small as possible. This value in part depends on the choice of $t\_0$. Larger $t\_0$ _typically_ ensures that we get smaller loss values (although this may not always be the case). However, also note that this creates a trade-off. If we set the $t\_0$ value to be too large (with respect to the total number of training iterations), it will in turn reduce the quality of our ensemble because there are fewer models in our moving average, or because the model could have converged (thereby reducing diversity along the optimization path). Finally, since we make use of pre-trained models in our work, we are able to use values of $t\_0$ close to initialization because the loss term reduces at a faster rate compared to a model being trained from scratch. This allows the moving average model to accumulate a larger number of model states along the optimization trajectory thereby helping reduce the variance term better.
> > >
> > > 2. Thank you for going into detail. We want to clarify that Spearman correlation is better when closer to 1 because we want to perform model selection based on the validation performance on an in-domain validation set, and we want this model to perform well on an out-domain test set. Clearly, if this is to hold true, we ideally want that the performance on the in-domain validation set and the out-domain test set are positive correlated along the optimization trajectory, and therefore close to 1. In regards to the outlier case of the Photo dataset where the correlation for the ensemble was negative, our explanation is that both the in-domain validation performance and out-domain test performance converge soon after training starts. And since both these values have converged, the rank correlation is largely noise because all the small differences in variations in both these curves get accumulated in the formula of Spearman correlation. This explanation is same as the one in our response before, but we have tried to clarify it further. We hope it is clearer now.

---

### Official Review · Reviewer_ibW9 · 2022-06-30

**Rating:** 6
**Confidence:** 4
**Soundness:** 3 good
**Presentation:** 3 good
**Contribution:** 2 fair

**Summary:**

This paper proposes two different models for OOD generalization, evaluated on the DomainBed benchmark across pretrained encoders and compared to representative baselines. First, it introduces a new hyperparameter free model averaging (MA) strategy during training called SMA, inspired by tail-averaging in optimization. SMA averages all parameters from a pre-defined starting point. Second, it proposes to ensemble several SMAs (EoA); EoA achieves SOTA results on DomainBed and mitigates instability in OOD performance across training.

In addition to these two models, it proposes a new model selection procedure for MA that performs early stopping and inference with SMA instead of the online model. This model selection procedure is justified by an extensive study of OOD performance stability throughout training and rank correlation between validation and test performance.

Finally, it explains the success of ensembling and MA using the Bias-Variance decomposition, adapted to a domain generalization setting.

**Questions:**

* How realistic is the theoretical analysis w.r.t. experimental setup?
    * the proposed bias-variance decomposition does not account for different seeds and hyperparameters used in the experiments.
    * l188 “the expected test domain error of an ensemble is strictly less than that of an individual model” + “that of ensembles is dominated by the bias term alone, and is thus strictly lower” l661: this analysis is valid only when the ensemble has infinite number of members. In practice, an ensemble has finite number of members, here 6 members for EoA, such that this analysis is unrealistic.
    * Authors mention that MA has less diversity than ensembles which consider different datasets, hyperparameters and seeds l678-679. This puts into question the validity of l195 “model averaging on the other hand has been shown to approximate an ensemble” and of the subsequent analysis.

* Unjustified statements in the theoretical section:
    * why does EoA outperform traditional ensembling theoretically?
    * “ensembles of larger size typically have better out-domain performance." l219-220; not explained by the analysis.
    * “$\hat{\theta}_T-\theta_t$ may not behave similar to that in their case” l207-208. Could you clarify this statement? How valid is the second order approximation of Izmailov2018 for tail-averaging if the behavior is not the same?
    * “the model averaging protocol used in our work behaves like an ensemble” l215-216: the outputs of the ensemble should be also plotted along the first and second terms to show that (or a distance between predictions). Here, only the first and second terms are plotted.

* Additional experiments:
    * How does SMA compare to SWA or SWAD in-domain? Section 4.2 compares only SMA to ERM. Cha2021 already showed that SWAD improved ERM in-domain so these conclusions are not new.
    * EoA should be evaluated on a same training / validation split to be fairer (cf limitation section).

* MIRO with RegNetY-16GF is reported at 74.1 and not. 70.1 in the original paper.

Cha2021: SWAD: Domain Generalization by Seeking Flat Minima


**Limitations:**

The limitations highlited by the authors are interesting. Yet, several important limitations are not enough discussed / missing:

* How realistic is the theoretical analysis w.r.t. experimental setup? C.f. question above.

* The SoTA results are obtained with two major limitations which are not discussed. This puts into question the fairness of the comparison.
    * high inference cost of EoA and ensembling: 6 times higher than the baselines; this important question is not discussed by the authors.
    * the ensembling and EoA models see more data than baselines as there are trained on different training/validation splits. It seems normal that the results are better than using a single training/validation split. A fairer evaluation would only consider a same training/validation split.


**Strengths And Weaknesses:**

Strengths:

* Ensembling MAs is novel to my knowledge and this model achieves SoTA results on the challenging DomainBed benchmark (yet, open question of fairness of comparison c.f. below). Authors performs thorough empirical validation (across datasets, architectures) and ablation studies. The stability of MA models w.r.t. online models OOD is interesting and well evaluated.
* SMA and the corresponding model selection procedure are not groundbreaking but have practical use: SMA removes hyperparameters in SWAD, the model selection procedure is more robust than in SWAD. It is thus more applicable than the original work SWAD that motivated this paper.
* It also proposes some theoretical results for explaining the success of ensembling and MA OOD by adapting the bias-variance decomposition to domain generalization (yet with several limitations detailed below).
* The paper is well written and easy to follow.

Weaknesses:

* Limited novelty of some contributions:
    * SMA is mostly based on related models (SWA and SWAD); it “simply” removes unnecessary hyperparameters but the principle of averaging models is unchanged. Ensembling SMAs is novel but this contribution is mostly empirical as a theoretical explanation of its success over standard ensembling is missing.
    * The change in the model selection procedure of SMA vs SWAD is also minor although useful in practice; it would be interesting to see if better stability with MA applies to other learning objectives (e.g. domain-invariance criterias) to better highlight this contribution.
    * The theoretical section simply applies the well-known bias-variance trade-off to domain generalization with minor consideration of the experimental setting (c.f. questions below). A big part of the theoretical results (Taylor expansion l198) tries to validate empirically an existing result (Izmailov2018). The novelty of this section would be improved if an explanation why EoA outperforms standard ensembling is provided, which is currently missing.

Two other weaknesses that justify my score:
* Question on applicability of theoretical results to practise and unjustified statements in the theoretical section (see questions below). Missing explanation of the sucess of EoA.

* Fairness of the comparison to baseline models (see limitations below). In particular, EoA and the ensemble baseline see more training data than other baselines.

Izmailov2018: Averaging wights leads to wider optima and better generalization.

---

> ### Author Response · Authors · 2022-07-30
> **Rebuttal Part 2/2**
>
> **EoA should be evaluated on a same training / validation split to be fairer (to single models)**:
> It does not make sense for each model in the EoA to be trained using the same training and validation split because of 2 reasons:
>
> 1. Since we use the same pre-trained model as initialization, aside from the impact of random hyper-parameter choices, there won't be much of a point of using an ensemble if all models are both initialized identically and trained on the same training set.
>
> 2. Importantly, by randomly splitting the training and validation sets for each model training in an ensemble, we are more closely simulating the theoretical bias-variance decomposition analysis because it involves an expectation over the training distribution.
>
> On a final note, the use of randomly chosen training-validation splits by ensembles should not be seen as an unfair advantage over a single model, but rather it should be seen as a limitation of non-ensemble models. This is because non-ensemble models have only one choice of splitting the training data into train/validation sets. Note that the two splits are still part of the training process even though the validation set is not used for gradient updates. The simplest way to remedy this limitation would be to use an ensemble, which is what we do.
>
> **$\\theta\_T - \\theta\_t$ may not behave similar to that in their case” Line 207-208. Could you clarify this statement? How valid is the second order approximation of Izmailov2018 for tail-averaging if the behavior is not the same?**:
>
> Clarification of the difference: The difference between the setting of Izmailov2018 (SWA) and our setting is that they apply SMA towards the end of training (last 25% of the total epochs), while we propose to apply it starting close to the beginning of training (starting after 100 iterations).
>
> Validity of the approximation: Note that the second order Taylor's expansion analysis discussed in our paper was not done in Izmailov2018. The validity of the second order approximation for our case is corroborated by the experiments in Fig 4 and Fig 9, where we show that the second order term concentrates around 0. This is the most direct verification of our second order Taylor's approximation analysis, which supports our claim that SMA roughly approximates the behavior of an ensemble of the mode states along the optimization trajectory.
>
> **How does SMA compare to SWAD in-domain?**:
> We could not compare SMA to the SWAD results for the in-domain setting because they did not report their experimental setup for this setting. In terms of comparison, we can expect SMA to perform similarly to SWAD in the in-domain setting while retaining the advantage of being a computationally cheaper alternative, as discussed and shown by the out-domain experiments.
>
> **MIRO with RegNetY-16GF is reported at 74.1 and not. 70.1 in the original paper**:
> Thank you for catching this typo.

---

> > ### Comment · Reviewer_ibW9 · 2022-08-03
> > **Thank you for your response**
> >
> > **Theoretical explanations**
> > Thank you for your clarifications, I better understood your theoretical analysis which motivates why EoA performs better than ensembling or single model. p6 would benefit from a reformulation to better understand the outcomes of your analysis, as it felt unclear in the original submission.
> >
> > **Fairness of experiments**
> > Are you fixing the same classifier’s initialization across runs? I do not fully agree as the ensemble would still benefit from different hyperparameters, batch ordering and initialization (if the classifier’s initialization varies).
> >
> > Anyways, the rebuttal helped fix my main concerns on the theoretical section. Therefore, I revised my score to weak accept; the proposed paper makes some interesting contributions with moderate impact.

---

> > > ### Author Response · Authors · 2022-08-04
> > > **Thank you**
> > >
> > > Thank you for your consideration.
> > >
> > > **Theoretical explanations**: We will update the text to make the above points clearer.
> > >
> > > **Fairness of experiments**: The classifier's initialization are not fixed across runs, only the deep network's initializations are. We agree with you that the ensemble will indeed benefit from different hyper-parameters and batch ordering, and we believe such differences between the different models in an ensemble help improve its performance, because if we remove them, then all the trained models in an ensemble will be identical to one another.

---

> ### Author Response · Authors · 2022-07-30
> **Rebuttal Part 1/2**
>
> Thank you for your comments. Please find our response below.
>
> **theoretical explanation for the success of EoA over standard ensembling is missing**:
> The bias-variance decomposition analysis discussed in our paper tells us that the expected test domain error of an ensemble is strictly less than the expected error of an individual model. This interpretation explains why a traditional ensemble of infinitely many unaveraged models can be expected to perform better than individual unaveraged models. However, this is infeasible. To mitigate this computational challenge, we propose EoA.
>
> To explain why EoA outperform traditional ensembles in practice, we note that we typically train a small number of independent models to form a traditional ensemble due to computational constraints. Thus traditional ensembles do not behave identically to the expected ensemble. Model averaging, which we show approximates an ensemble of models along the optimization trajectory, due to the Taylor's approximation, applied along side ensembling behaves more similarly to the expected ensemble function. This is our explanation for why EoA outperform traditional ensembles in practice. This is also corroborated by Fig 3 where we show that for increasing ensemble size, EoA consistently outperforms traditional ensembles.
>
> To be clear, note that we are not claiming that EoA of a finite number of models behaves exactly as the expected ensemble. We are rather claiming that its behavior is closer to the expected ensemble compared to a traditional ensemble of the same size.
>
> We discussed the above theoretical reason in Section 3.2 Line 191 onwards.
>
> **the proposed bias-variance decomposition does not account for different seeds and hyperparameters used in the experiments**:
> This is not true. The bias-variance decomposition equation does account for the different seeds. Since all models in an ensemble are initialized using the same pre-trained model, it is the randomness in seed that decides the split of the training domains into training and validation subsets (following Gulrajani et al 2020), which is what the expectation in the bias variance decomposition is over.
>
> You are correct in pointing out that the decomposition does not account for the different hyper-parameters. However, one can trivially add an expectation term over hyper-parameters in the bias-variance decomposition equation without changing our conclusion (since individual models would still be bound to use just a single sample of the set of hyper-parameters).
>
> **In practice, an ensemble has finite number of members, here 6 members for EoA, such that this (bias-variance) analysis is unrealistic**:
> We discuss precisely this on Line 192-193, and discuss how the use of SMA as a part of EoA relaxes this assumption and makes our analysis realistic. We have also explained this above in our response to the 1st question.
>
> **MA has less diversity than ensembles…This puts into question the validity of Line 195 “model averaging on the other hand has been shown to approximate an ensemble”**:
> If you kindly look at our statements without bias, it should be apparent that we are claiming that MA has less diversity compared to true ensembles, however, using an ensemble of MA models is a middle ground between using a single MA model (less diversity) and using a very large number of un-averaged models in a traditional ensemble (large diversity), while being computationally cheaper compared to the latter.
>
> **“ensembles of larger size typically have better out-domain performance." Line 219-220; not explained by the analysis**:
> Theoretically, our analysis explains this because:
>
> 1. larger number of models in an ensemble better approximate the expected ensemble behavior (Monte Carlo approximation), and
>
> 2. the bias variance decomposition analysis shows that the expected ensemble model has lower test domain error compared to the expected test domain error of a single model.
>
> Figure 3 further empirically corroborates this claim by showing that ensembles of a larger size has better out-domain performance.

---

### Official Review · Reviewer_UuXX · 2022-07-09

**Rating:** 6
**Confidence:** 5
**Soundness:** 3 good
**Presentation:** 3 good
**Contribution:** 3 good

**Summary:**

This paper shows that simple model averaging over the course of training improves the rank correlation between in-domain validation accuracy and out-of-domain test accuracy. They also show that ensembling these averaged models further improves this rank correlation, more so than ensembling unaveraged models. Finally, they illustrate that this improved rank correlation translates to improved performance on DomainBed, given how crucial it is for model selection or early stopping in domain generalization.

**Questions:**

- Does SMA, or learning rate decay, reliably improve other DG methods? E.g. IRM, VREx, CORAL, etc.
- If the model is trained from scratch without pre-training, would you expect a larger t_0 to be needed? How would you select this hyperparameter?
- Possible to move Table 7/8/9 into the main paper? In my opinion, these are the most insightful results.
- [Minor] Why do the authors choose the notation $\hat{\theta}_t$ over say $\bar{\theta}_t$ for the averaged parameters?
- [Minor, comment] Table 4: clarify in caption that training-domain validation set was used for model selection.


**Limitations:**

Yes, in Appendix E.

**Strengths And Weaknesses:**

**Strengths:**
- *Clear, well-written paper.*
- *Comprehensive and insightful experiments:* There has been a recent trend of cherry-picking DG datasets, so I appreciated the wideness of the evaluation here, with only CMNIST and RMNIST left out of DomainBed (understandably so).
- *Simple and effective:* Averaging models, or simply decaying the learning rate (see below), is a very simple idea. Despite being well-explored outside of DG, its effectiveness for model selection in DG is quite surprising, and should perhaps become the standard in DG benchmarks where reliable model selection hinders fair comparison.

**Weaknesses:**
- *The simple moving average (SMA) is just learning rate decay and this connection is not made.*
    - Let $\theta_t = \hat{\theta}\_{t-1} - \eta \nabla_{\theta} L$, with $\eta$ the learning rate, and $t_0=0$. Then we can rewrite Eq. 1 of the paper as:
\\begin{align}
\\hat{\\theta}\_t &=  \\frac{t}{t +1} \\cdot \\hat{\\theta}\_{t-1} + \\frac{1}{t + 1} \\cdot \\theta_t \\\\
&= \\frac{t}{t +1} \\cdot \\hat{\\theta}\_{t-1} + \\frac{1}{t + 1} \\cdot (\\hat{\\theta}\_{t-1} - \\eta \\nabla_{\\theta} L ) \\\\
&= \\hat{\\theta}\_{t-1} - \\frac{\\eta}{t + 1} \\nabla_{\\theta} L \\\\
\\end{align}
    - Thus, a simple moving average of model weights is equivalent to a linear learning rate decay.
    - Adding back in t_0 allows a standard warm-up period before decaying the learning rate, and corresponds to tail averaging.
    - Indeed, the stochastic weight averaging paper (SWA, [1]) discussed the importance of learning rate scheduling, and in fact they decay the learning rate for the first 75% of training, then set the learning rate to a high constant value for the remaining 25% of the time in order to increase the diversity of the solutions / optima found. This paper essentially removes this diversity-seeking trick, which is also employed in the DG follow-up paper SWAD [2], and reverts to decaying the learning rate until the end of training. The diversity for model ensembling then comes from different initializations.
    - *Main issue:* Model averaging, and “ensembles of averages”, takes center stage in this paper. However, it just translates to using linear learning rate decay when ensembling, and this connection is never made.
    - *Potential remedies:*
      - Make this connection clear throughout the paper, particularly in Section 2 and in the related work section. This could include a discussion of how SWA and SWAD deliberately avoid learning rate decay in the tails in order to arrive at more diverse solutions, which is perhaps undesirable for DG model selection.
      - An experiment where linear learning rate decay is employed instead of model averaging, as this should be equivalent.
      - Discuss works which explored the benefits of learning rate decay.
    - I will increase my score if this can be adequately addressed.
- *Not exactly hyperparameter-free, as claimed:*
    - Throughout the paper, it is stressed that the proposed approach is hyperparameter free, and that this is a major advantage over SWAD.
    - However, the method still requires specification of a start iteration t_0 and an averaging frequency. For t_0, the authors “arbitrarily choose 100 as the start iteration”. While Appendix C demonstrates that the proposed method is not too sensitive to these values for the datasets and pretrained models explored in the paper, they likely still need to be selected for new datasets and models, particularly if the models are trained from scratch.
    - *Remedy:* tone down the hyperparameter free claim/statement, particularly in the contribution list of the introduction. Can still say it is much less sensitive than SWAD, which it appears to be.

[1] Izmailov, P., Podoprikhin, D., Garipov, T., Vetrov, D., & Wilson, A. G. (2018). Averaging weights leads to wider optima and better generalization. In 34th Conference on Uncertainty in Artificial Intelligence (pp. 876-885).

[2] Cha, J., Chun, S., Lee, K., Cho, H. C., Park, S., Lee, Y., & Park, S. (2021). SWAD: Domain generalization by seeking flat minima. In Advances in Neural Information Processing Systems, 34, (pp. 22405-22418).

---

> ### Author Response · Authors · 2022-07-30
> **Thank you for your comments. Please find our response below.**
>
> **The simple moving average (SMA) is just learning rate decay and this connection is not made**:
> This is not true. The derivation you showed has a problem in the assumption, i.e.,
>
> $\\theta\_t = \\hat{\\theta}\_{t-1} - \\eta \\nabla\\theta L$.
>
> Our explanation is as follows. Under the gradient descent update rule,
>
> $\\theta\_t = {\\theta\_{t-1}} - \\eta \\nabla\_\\theta L$
>
> Notice the first term in R.H.S. is $\\theta\_{t-1}$ and not $\\hat{\\theta}\_{t-1}$. These two terms have different values since $\\theta\_{t-1}$ is achieved by taking a gradient descent step from $\\theta\_{t-2}$ in the direction of $\\nabla\_{\\theta\_{t-2}} L$, while $\\hat{\\theta}\_{t-1}$ is the average of $\\theta\_0, \\theta\_1,...,\\theta\_{t-1}$.
>
> To more concretely see that we cannot replace SMA with learning rate decay in general, let's consider the simplest non-trivial case where we do just 2 gradient descent updates (T=2):
> 1. Case 1: using a fixed learning rate $\eta$ and then compute the SMA at T=2. This is the traditional SMA approach.
> 2. Case 2: using learning rate decay using the equation you proposed
>
> We can then compare the two and check if they are identical. These are shown below:
>
> 1. Case 1:
>
> $\\theta\_1 = \\theta\_0 - \\eta \\nabla\_{\\theta\_0}L$
>
> $\\theta\_2 = \\theta\_1 - \\eta \\nabla\_{\\theta\_1}L$
>
> Thus,
>
> $\\hat{\\theta}\_2 = 1/3.(\\theta\_2 + \\theta\_1 + \\theta\_0)$
>
> After substitution, we get,
>
> $\\hat{\\theta}\_2 =\\theta\_0 - 2\\eta/3 \\nabla\_{\\theta\_0}L - \\eta/3 \\nabla\_{\\theta\_1}L $
>
> 2. Case 2:
>
> $\\theta\_1 = \\theta\_0 - \\eta/(1+1) \\nabla\_{\\theta\_0}L$
>
> $\\theta\_2= \\theta\_1 - \\eta/(2+1) \\nabla\_{\\theta\_1}L$
>
> After substitution, we get,
>
> $\\theta\_2= \\theta\_0 -\\eta/2 \\nabla\_{\\theta\_0}L - \\eta/3 \\nabla\_{\\theta\_1}L$
>
> By your definition, $\\theta\_2$ is the same as $\\hat{\\theta}\_2$ in Case 2 since we used learning rate decay. But as you can see, $\\hat{\\theta}\_2$ from case 1 and case 2 do not agree, and the difference between then two cases will increase as we make T larger. Hence SMA is not equivalent to learning rate decay.
>
> **Indeed, the stochastic weight averaging paper (SWA, [1]) discussed the importance of learning rate scheduling…The diversity for model ensembling then comes from different initializations**:
> As explained above, since a simple moving average is not equivalent to learning rate decay, our proposal does not decay the learning rate as training progresses. Even the explicit learning rate in the Adam optimizer is set to a constant value throughout the training in our experiments, which is also the case in SWAD.
>
> The statement "diversity for model ensembling then comes from different initializations" is not true, because in each experimental setup in our paper, we use the same pre-trained model state as our initialization for all the models in an ensemble.
>
> **I will increase my score if this can be adequately addressed**:
> Thank you for your unbiased approach, we appreciate it. We hope that our analysis above addresses your concern regarding the equivalence of SMA and learning rate decay.
>
> **it is stressed that the proposed approach is hyperparameter free…the method still requires specification of a start iteration t_0 and an averaging frequency**:
> We will tone down the hyper-parameter-free claim and mention that the SMA hyper-parameters are much less sensitive compared to SWAD, per your suggestion.
>
> **Does SMA, or learning rate decay, reliably improve other DG methods?**:
> As shown in the SWAD paper, combining moving average models with existing domain generalization (DG) methods does show improvements. Since our proposal is essentially different from SWAD in that it is computationally more efficient, we expect the same conclusion should apply in regards to performance improvements upon combination with existing DG methods.
>
> **If the model is trained from scratch without pre-training, would you expect a larger t_0 to be needed?**:
> For models trained from scratch, it is typical in literature to either use exponential moving average or use SWA. Our proposal of SMA is specific to the case when we initialize the model using a pre-trained model, in which case we found that t_0 near this initialization works better. Note that our other proposal of using an ensemble of averaged models (which can be combined with either exponential or SWA) can be expected to improve performance over individual models based on our Bias-Variance theoretical analysis.
>
> **Why do the authors choose the notation $\\hat{\\theta}_t$ over $\\bar{\\theta}_t$?**:
> It was an arbitrary choice.
>
> **Table 4: clarify in caption that training-domain validation set was used for model selection**:
> Added it.

---

> > ### Comment · Reviewer_UuXX · 2022-08-03
> > **Thank you for your response**
> >
> > **Learning rate decay:** Thank you for this clarification -- the initial assumption was indeed a mistake on my part! I now see that SMA and linear learning rate decay are not equivalent in general, and should perhaps have toned down my initial claim/language.
> >
> > **Learning rate scheduling in SWA and SWAD:** While SMA and learning rate decay are indeed not equivalent, and diversity does indeed not come from initialisation due to the use of the same pretrained model, I think there is still a valid point in this paragraph: SWA and SWAD explicitly seek diversity by increasing the learning rate at the end of training, while your method does not do so, and in fact explicitly seeks less diversity (or further "stability") through a SMA of the weights. This is evidently beneficial for model selection in DG, and seemed like an important difference with SWAD that was worth explicitly pointing out.
> >
> > **"Our proposal of SMA is specific to the case when we initialize the model using a pre-trained model":** I see, I did not realize this.

---

> > > ### Author Response · Authors · 2022-08-04
> > > **Thank you**
> > >
> > > Thank you for your consideration.

---

### Official Review · Reviewer_ao9C · 2022-07-13

**Rating:** 7
**Confidence:** 4
**Soundness:** 3 good
**Presentation:** 3 good
**Contribution:** 3 good

**Summary:**

Authors point out the chaotic behavior exists even along the training optimization trajectory of a single  model, and propose a simple model averaging protocol that both significantly boosts domain generalization performance. Authors also show that instead of simple moving average, ensembling moving average models from multiple runs can boost the performance even further. Authors show results on various popular datasets and show the effectiveness of proposed method.

**Questions:**

1) SMA - Line # 54 SMA abbreviation is used before actually defining it.

2) Results on datasets: Take an example of PACS. Authors show average results on PACS dataset. But in appendix results on per domain show substantially worse results on Photos. Is it surprising or expected? Some discussion on this will be really helpful.

3) What is the time complexity difference between SMA and EoA? It looks like EoA may have much higher complexity but performance gains are not much over SMA. Can authors comment?

4) Would using SMA or EoA on top of some existing methods (e.g. MIRO) improve the performance further? Is there anything that stops us from combining SMA with other methods too?

5) How would one scale the current approach when number of domains are really large? Do authors see any issue in this?

6) Right y-axis of Figure 1: "trainig" → training



**Strengths And Weaknesses:**

Strengths

1) Figure 1 explains the motivation and intuition of what authors want to do.
2) Paper is clearly written and even though idea is simple, authors show the effectiveness of the proposed method.


Weaknesses
1) Complexity and run times compared to existing methods should be discussed
2) Summary of limitations should be discussed in the paper

---

> ### Author Response · Authors · 2022-07-30
> **Thank you for your comments. Please find our response below.**
>
> **What is the time complexity difference between SMA and EoA?; Complexity compared to existing methods**:
> Since EoA trains an ensemble of models, the complexity scales linearly with the number of models used in the ensemble compared with SMA on a single model. Of course, these different models can be trained in parallel if resources are available, given each model is trained independently of one another. Finally, the computational overhead due to SMA is practically negligible (compared to backprop) since it merely involves a running average estimate of the parameters. So its complexity is the same as that of training a vanilla supervised deep network.
>
> **Would using SMA or EoA on top of some existing methods (e.g. MIRO) improve the performance further?**:
> As shown in the SWAD paper, combining moving average models with existing domain generalization (DG) methods does show improvements. Since our SMA proposal is essentially different from SWAD in that it is computationally more efficient, we think the same conclusion should apply in regards to performance improvements upon combination with existing DG methods. But EoA on the other hand can be expected to further improve performance.
>
> **How would one scale the current approach when number of domains are really large?**:
> We performed preliminary experiments in which we stochastically pick one of the training domains at every iteration, and sampled a mini-batch only from that domain to update parameters as opposed to using samples from all the domains. We found that this protocol resulted in a similar performance as that achieved by the protocol used in our work. This addresses the memory issues one would face in cases where the number of domains is large. This issue is discussed in the Discussions section in appendix E under the heading Scalability in the submitted paper on page 18.
>
> **...PACS dataset…in appendix results on per domain show substantially worse results on Photos. Is it surprising or expected?**:
> Were you referring to another table? The EoA accuracy on the Photos domain of the PACS dataset is consistently better compared to all compared methods for all the 3 architectures used in our experiments.
>
> **Summary of limitations should be discussed in the paper**:
> We will try to summarize the limitations (section E in appendix) in the main text if additional space is allowed.
>
> **SMA - Line # 54 SMA abbreviation is used before actually defining it.**:
> The SMA abbreviation is defined on Line 45.

---

### Meta-Review · Area_Chair_JudE · 2022-08-27

**Recommendation:** Accept
**Confidence:** Less certain

**Metareview:**

This work introduces a hyperparameter-free model averaging and ensembling scheme for the domain generalization setting that achieves state-of-the-art results. Though the proposed method is simple and is related to existing techniques, the strong and comprehensive evaluation demonstrates that it is highly effective and therefore is of interest to the community. In addition, the paper is clear and well written, with careful analyses and useful theoretical intuitions. In summary, this paper will be a useful addition to the NeurIPS program.

**Award:**

No

---

### Decision · Program_Chairs · 2022-09-14

Accept